# HDAC3: A Multifaceted Modulator in Immunotherapy Sensitization

**DOI:** 10.3390/vaccines13020182

**Published:** 2025-02-13

**Authors:** Rui Han, Yujun Luo, Jingdong Gao, Huiling Zhou, Yuqian Wang, Jiaojiao Chen, Guoyin Zheng, Changquan Ling

**Affiliations:** 1Oncology Department of Chinese Medicine, The First Affiliated Hospital of Naval Medical University, Shanghai 200433, China; ruihan@smmu.edu.cn (R.H.);; 2Department of Chinese Medicine, Naval Medical University, Shanghai 200433, China; 3Oncology Department, Suzhou TCM Hospital Affiliated to Nanjing University of Chinese Medicine Suzhou, Suzhou 215009, China

**Keywords:** HDAC3, tumor immunotherapy, immune modulation, combination therapy, tumor microenvironment

## Abstract

Histone deacetylase 3 (HDAC3) has emerged as a critical epigenetic regulator in tumor progression and immune modulation, positioning it as a promising target for enhancing cancer immunotherapy. This work comprehensively explores HDAC3’s multifaceted roles, focusing on its regulation of key immune-modulatory pathways such as cGAS-STING, ferroptosis, and the Nrf2/HO-1 axis. These pathways are central to tumor immune evasion, antigen presentation, and immune cell activation. Additionally, the distinct effects of HDAC3 on various immune cell types—including its role in enhancing T cell activation, restoring NK cell cytotoxicity, promoting dendritic cell maturation, and modulating macrophage polarization—are thoroughly examined. These findings underscore HDAC3’s capacity to reshape the tumor immune microenvironment, converting immunologically “cold tumors” into “hot tumors” and thereby increasing their responsiveness to immunotherapy. The therapeutic potential of HDAC3 inhibitors is highlighted, both as standalone agents and in combination with immune checkpoint inhibitors, to overcome resistance and improve treatment efficacy. Innovative strategies, such as the development of selective HDAC3 inhibitors, advanced nano-delivery systems, and integration with photodynamic or photothermal therapies, are proposed to enhance treatment precision and minimize toxicity. By addressing challenges such as toxicity, patient heterogeneity, and resistance mechanisms, this study provides a forward-looking perspective on the clinical application of HDAC3 inhibitors. It highlights its significant potential in personalized cancer immunotherapy, paving the way for more effective treatments and improved outcomes for cancer patients.

## 1. Introduction

HDAC3 (histone deacetylase 3) is an enzyme that plays a crucial role in regulating gene expression by altering the acetylation status of histones, the proteins that DNA is wrapped around. By removing acetyl groups from histones, HDAC3 leads to a more condensed chromatin structure, which restricts access to the DNA for transcription machinery and thus suppresses gene transcription [1,2]. In addition to its function in chromatin remodeling, HDAC3 is involved in key cellular processes such as cell cycle regulation, DNA repair, and apoptosis [3]. Existing evidence indicates that HDACs play an essential role in the progression of various diseases. One study demonstrated that a series of novel single-pyrene-derived hydroxyl acids significantly inhibit HDAC activity, thereby reducing the production of Aβ and decreasing tau protein phosphorylation, which helps restore cognitive function impaired by Alzheimer’s disease. Based on their physiological and pathological effects, the HDAC family can be classified into four categories: I, II, III, and IV. Notably, the HDAC family often exhibits abnormally high expression in tumors, promoting tumor growth, invasion, and metastasis. Various types of HDAC inhibitors are currently under gradual development, with a particular focus on exploring and developing selective HDAC inhibitors to optimize the absorption, distribution, metabolism, and excretion characteristics of these compounds while minimizing toxicity and side effects [4,5,6]. As a member of the class I histone deacetylase family, HDAC3 operates as part of larger multi-protein complexes, such as the SMRT (silencing mediator of retinoid and thyroid receptors) and NCoR (nuclear receptor corepressor) complexes, which are essential for transcriptional repression [7,8]. Although HDAC1, HDAC2, and HDAC3 are all members of the Class I HDAC family, they perform distinct functions in gene regulation and expression. HDAC1 and HDAC2 team up with NURD, SIN3, and CoREST to form transcriptional repression complexes. They have different expression patterns and play a regulatory role in neurodevelopment and synaptic plasticity, with higher expression in actively proliferating cells. On the contrary, HDAC3 forms inhibitory complexes with NCOR or SMRT. These complexes are widely present in metabolic tissues and are crucial for regulating and reshaping metabolic activities and inflammatory responses. Since tumor occurrence is closely linked to abnormal metabolic activities and unpredictable strong inflammatory responses, HDAC3 has attracted increasing exploration and attention as a target for tumor treatment strategies [9]. In particular, our published study has primarily investigated the multiple anti-cancer effects of HDAC3 inhibition treatment, which also shows great potential in enhancing the sensitivity of cancer immunotherapy [10,11]. Its enzymatic activity is dependent on interactions with these corepressors and associated proteins, further highlighting its importance in regulating cellular functions and tumor progression [12,13,14].

HDAC3 has been increasingly recognized for its role in cancer development and progression [15]. In many types of tumors, such as gastric cancer, glioma, and breast cancer, HDAC3 is often overexpressed, and its elevated levels are associated with more aggressive disease characteristics and poor patient outcomes. HDAC3 contributes to cancer by promoting the proliferation, survival, and metastasis of tumor cells through the regulation of gene expression [16,17]. One of the primary mechanisms by which HDAC3 facilitates tumor growth is by repressing the expression of genes involved in tumor suppression, apoptosis, and cell cycle control [18]. By removing acetyl groups from histones, HDAC3 causes chromatin to adopt a more condensed state, preventing the transcription of these critical regulatory genes [19,20]. This leads to unchecked cell division, evasion of apoptosis, and enhanced metastatic potential. Furthermore, HDAC3’s role extends beyond histone modification. It also affects non-histone proteins that are involved in key signaling pathways, further contributing to oncogenesis [21,22]. For instance, HDAC3 can protect proteins like POC5 (POC5 centriolar protein), which supports cancer cell proliferation, from degradation [23]. Additionally, studies have shown that high HDAC3 expression correlates with resistance to certain therapies, making it a potential target for cancer treatment [24,25,26].

Given its influence on tumor biology, HDAC3 is being studied as a potential biomarker for cancer prognosis, as well as a therapeutic target. HDAC3 inhibitors are being explored in clinical trials to disrupt its activity, aiming to slow tumor growth and improve patient outcomes, either alone or in combination with other cancer treatments [27]. One of the earliest HDAC inhibitors approved for cancer treatment is vorinostat (SAHA), which targets multiple HDAC isoforms, including HDAC3 [28]. Other broad-spectrum inhibitors like romidepsin and panobinostat have also shown efficacy in treating hematological malignancies [29]. More recently, selective HDAC3 inhibitors have garnered attention for their potentially improved efficacy and reduced toxicity compared to pan-HDAC inhibitors [30,31]. These include compounds such as RGFP966, which has demonstrated promising preclinical results in models of cancer, exhibiting selective inhibition of HDAC3 activity while sparing other HDAC isoforms [32]. Combination therapies that include HDAC3 inhibitors with other treatment modalities such as chemotherapy, radiation, or immunotherapy are being explored to enhance therapeutic outcomes [33]. HDAC3 inhibitors, when combined with DNA-damaging agents or immune checkpoint inhibitors, can sensitize tumor cells to these treatments, leading to more effective cancer cell eradication [34,35]. Studies have demonstrated that HDAC3 inhibition can enhance the efficacy of PARP inhibitors in breast and ovarian cancers by disrupting DNA repair mechanisms, rendering cancer cells more vulnerable to DNA damage [36]. Additionally, HDAC3 inhibitors have shown potential in overcoming drug resistance. For instance, in hormone receptor-positive breast cancer, HDAC3 inhibition can restore sensitivity to endocrine therapies like tamoxifen, suggesting a role for HDAC3 inhibitors in tackling therapeutic resistance [37].

In recent years, tumor immunotherapy has made significant strides, particularly through immune checkpoint inhibitors (ICIs) that target pathways like PD-1/PD-L1 and CTLA-4 [38]. These therapies have revolutionized cancer treatment, offering long-term survival benefits for certain patients, especially in cases of melanoma, lung cancer, and other solid tumors [39,40,41]. Despite these advances, a substantial proportion of patients do not respond to immunotherapy, or they eventually develop resistance after an initial response [42]. This challenge highlights the urgent need for sensitizers to enhance the efficacy of immunotherapy. These sensitizers could potentially modify the tumor microenvironment, enhance antigen presentation, or improve immune cell infiltration to overcome resistance mechanisms [43,44]. Combining immunotherapy with agents that increase tumor immunogenicity or reverse immune suppression is crucial to expanding the treatment’s success to a broader range of patients. Identifying and developing such sensitizers is a critical area of ongoing research in order to maximize the therapeutic potential of immunotherapy and provide more effective treatment options for cancer patients [45]. This continuous exploration of immune-modulating agents not only seeks to improve response rates but also addresses the pressing need to manage resistance in patients who might otherwise have limited options.

The rationale behind targeting HDAC3 for tumor immunotherapy sensitization lies in its ability to modulate the immune landscape. Inhibition of HDAC3 can promote the expression of genes related to antigen presentation and immune recognition, thereby enhancing the visibility of tumor cells to the immune system. Preclinical studies have demonstrated that HDAC3 inhibitors can increase the efficacy of immune checkpoint inhibitors by reversing immune suppression and promoting a more robust anti-tumor immune response based on their effects on the tumor microenvironment by regulating immune cell infiltration and immune evasion mechanisms, which are critical for the success of immunotherapy [46].

Here, we have summarized the latest research on HDAC3 regulation of anti-tumor immunity to highlight the importance of HDAC3 as a therapeutic target for sensitizing tumor immunotherapy and its relevance in this research field, thereby encouraging more attention and investment in this area. Therefore, the combination of HDAC3 regulators with immunotherapy is proposed as an innovative tumor treatment strategy with significant clinical potential and research prospects. This approach offers a promising opportunity to enhance the overall prognosis for cancer patients.

## 2. Clinical Perspectives on the Roles of HDAC3 in Cancer Treatment

Currently, the commonly used clinical HDAC regulators are primarily multi-target HDAC inhibitors, which target HDAC1, HDAC2, HDAC3, and HDAC6 [47]. Several HDAC inhibitors have been approved for clinical use, primarily targeting hematologic malignancies. These include vorinostat, romidepsin, belinostat, and panobinostat, which are used in the treatment of peripheral T-cell lymphoma, cutaneous T-cell lymphoma, and multiple myeloma. Additionally, chidamide, approved in China, is used for the treatment of peripheral T-cell lymphoma and breast cancer [48,49,50,51,52]. We summarize the past five years of clinical research on the combination of HDACi and immunotherapy, as well as preclinical research on selective HDAC3i (Appendix A). Clinical trial NCT04233294 studies the effects of chidamide in combination with immunotherapy. The results indicated that the objective response rate of chidamide combined with decitabine and camrelizumab (CDP) was 94%, in comparison to historical controls. Notably, all patients who had previously shown resistance to treatment with drug-resistant decitabine and camrelizumab exhibited a therapeutic response following treatment with CDP. These inhibitors are predominantly pan-HDAC inhibitors, affecting multiple HDAC isoforms rather than being specific to HDAC3 [53]. Another clinical study, NCT04631029, evaluated the safety of the class I HDAC inhibitor entinostat in combination with standard doses of carboplatin, etoposide, and atezolizumab in patients with previously untreated extensive-stage small cell lung cancer (SCLC). A total of three patients were treated with entinostat at a dose level of 2 mg (DL1). The results indicated that serious adverse reactions occurred, including anemia, neutropenia thrombocytopenia, leukopenia, and hypocalcemia. During the first treatment cycle, two patients experienced dose-limiting toxicities (DLTs): one patient exhibited grade 4 febrile neutropenia, while the other experienced grade 5 sepsis. Based on a Bayesian optimal interval (BOIN) design, patient recruitment was halted, and the clinical trial was closed.

One possible speculation is that the application of multi-target HDAC inhibitors broadens the range of intervention targets, thereby activating more waterfall effects and side effects. However, replacing multi-target HDAC inhibitors with HDAC3 single-target inhibitors in the combination strategy could potentially optimize or avoid this issue (Figure 1).

HDAC3 single-target inhibitors may possess significant advantages over multi-target HDAC inhibitors, primarily in terms of specificity, toxicity management, resistance reduction, and therapeutic precision. By specifically targeting the HDAC3 isoform, these inhibitors regulate pathological pathways associated with HDAC3, achieving more precise anticancer effects [54,55,56]. This specificity not only effectively minimizes non-specific inhibition of other HDAC isoforms—thereby significantly reducing common hematologic and gastrointestinal adverse effects—but also decreases the risk of resistance caused by compensatory mechanisms triggered by broad inhibition [57,58]. Furthermore, since HDAC3 plays a pivotal role in various cancers, targeting HDAC3 allows single-target inhibitors to achieve superior efficacy in tumors driven by HDAC3 activity [59,60]. In combination with immunotherapy, HDAC3 single-target inhibitors still hold potential advantages over pan-HDAC inhibitors. First, through their precise mechanisms of action, HDAC3 single-target inhibitors can avoid overlapping toxicities with immunotherapy agents, thereby improving the tolerability and safety of combination treatments. Second, by sparing other HDAC isoforms, these inhibitors help maintain the normal balance of the immune system, preventing immune suppression often seen with non-specific inhibition; this is crucial for enhancing the efficacy of immunotherapy [61]. Additionally, HDAC3-specific inhibitors are more likely to synergize with immunotherapies, as they can modulate the tumor microenvironment and immune signaling pathways to further amplify antitumor immune responses, ultimately improving overall therapeutic outcomes. An experiment based on non-Hodgkin lymphoma diffuse large B-cell lymphoma (DLBCL) cells revealed that the resistance of DLBCL cells to pan-HDACi may be related to the failure to effectively inhibit HDAC3, and the use of selective HDAC3i can effectively reverse the resistance of DLBCL to pan-HDACi therapy [62].

Thus, HDAC3 single-target inhibitors show substantial promise in both monotherapy and combination immunotherapy, representing a critical direction in the development of future anticancer therapies. However, currently, no HDAC3-specific inhibitors have been approved for combination immunotherapy in the clinical treatment of cancer, leaving enormous room for research in this field.

Additionally, HDAC3 expression level has been considered to act as a potential indicator for prognosis [63]. Evidence has reported that high expression of HDAC3 in tumor tissues of glioma patients is negatively correlated with overall survival (OS) and relapse-free survival (RFS) [64]. Additionally, based on an analysis of six HDAC genes (HDAC1, HDAC3, HDAC4, HDAC5, HDAC7, and HDAC9), a prognostic model for glioma patients revealed that glioma cells with low expression of HDAC3 exhibited significantly reduced growth and invasive capabilities [65]. HDAC3 can also protect the centrosomal protein POC5 from proteasomal degradation by transactivating CREB1, thereby promoting the proliferation and metastasis of breast cancer cells both in vitro and in vivo [23]. Thus, the expression of HDAC3 has been considered to act as an indicator to predict the overall tumor recurrence, metastasis, and prognosis of patients [66,67,68,69].

## 3. HDAC3 and Its Effects on Immune Regulation

The tumor immune microenvironment (TIME) represents a dynamic ecosystem comprising tumor cells, immune cells, stromal elements, and soluble factors that collectively shape the immune response to cancer [70]. Key components of anti-cancer immunity include various immune cells such as cytotoxic T cells (CD8+), which target and kill tumor cells, and helper T cells (CD4+), which assist in activating and regulating immune responses. B cells are responsible for producing antibodies that can bind to tumor antigens, while natural killer (NK) cells provide rapid responses against stressed or transformed cells. Dendritic cells play a critical role as antigen-presenting cells, linking innate and adaptive immunity by activating T cells. Cytokines and chemokines serve as signaling molecules that modulate the behavior of these immune cells, enhancing their infiltration and function within the TIME. However, the presence of regulatory mechanisms, including regulatory T cells (Tregs) and myeloid-derived suppressor cells (MDSCs), often leads to an immunosuppressive microenvironment that facilitates tumor evasion from immune detection [71]. Histone deacetylase 3 (HDAC3) is an important factor influencing anti-cancer immunity, as it regulates gene expression linked to immune cell activation and differentiation [72,73]. HDAC3 can modulate the activity of T cells and macrophages, impacting their anti-tumor functions. Additionally, it may play a role in promoting an immunosuppressive environment by enhancing the expression of immune checkpoint proteins, which can limit effective immune responses [74,75,76,77,78]. Targeting HDAC3 could therefore represent a promising strategy to enhance anti-cancer immunity by reprogramming the TIME and overcoming immune evasion mechanisms employed by tumors.

### 3.1. HDAC3 and Immune Signaling Pathways

#### 3.1.1. The cGAS-STING Pathway and Its Immune Regulation

The cGAS-STING pathway is regarded as a central node in the regulation of both innate and intrinsic immunity, with its expression being influenced by HDAC3. As a crucial effector pathway in innate immunity, the cGAS-STING pathway exhibits a bi-directional regulatory role in the immune response. On one hand, it activates downstream pathways through the recognition of intracellular cyclic dinucleotides (CDN) and damage-associated molecules, promoting the release of substantial quantities of IFN-γ, chemokines, and inflammatory factors [79].

Activation of the cGAS-STING pathway upregulates the pro-apoptotic protein BAX while simultaneously reducing the expression of the anti-apoptotic protein BCL2 in tumor cells. This process results in the permeabilization of the outer mitochondrial membrane, which induces the expression of the caspase-3 gene, ultimately leading to the apoptosis of tumor cells [80,81]. Activation of the STING pathway facilitates the release of type I interferons. This activation enhances the cross-presentation capability of dendritic cells (DCs), which in turn activates tumor-specific CD8+ T cells and improves their lymph node homing ability by increasing the expression of CCR7 on DCs [82,83,84,85,86]. Collectively, these mechanisms contribute to biological self-protection. Tumor treatment strategies based on the cGAS-STING pathway are gradually being developed, and increasing evidence shows that cGAS-STING agonists can effectively eliminate tumors and induce durable anti-tumor immune responses when combined with immunotherapy [87,88].

#### 3.1.2. Regulation of the cGAS-STING Pathway by HDAC3

Evidence has shown that the activation level of the cGAS-STING pathway is regulated by HDAC3. For instance, in a mouse model with neuroinflammation, damaged mitochondrial DNA in the cytoplasm of microglia effectively activated the cGAS-STING pathway, thereby promoting the formation of an inflammatory microenvironment. In this context, HDAC3 transcription was found to enhance the expression of cyclic GMP-AMP synthase (cGAS). Inhibition of HDAC3 can impede the activation of the cGAS-STING pathway by obstructing the activation of the p65-cGAS-STING axis [89]. Additionally, in a mouse model of severe acute pancreatitis (SAP), silencing HDAC3 has been demonstrated to obstruct the activation of the cGAS-STING pathway in intestinal epithelial cells, inhibit oxidative stress and inflammation in these cells, and consequently enhance intestinal barrier function [90]. In an acute lung injury (ALI) mouse model, lipopolysaccharide (LPS) recruits HDAC3 to the promoter of the miR-4767 gene, thereby promoting cGAS expression in macrophages [91]. Therefore, HDAC3 may function as a positive activator of the cGAS-STING pathway. Inhibiting HDAC3 expression can partially impede the activation of the cGAS-STING pathway, thereby enhancing the feedback regulation of this pathway.

#### 3.1.3. Ferroptosis in Anti-Tumor Immunity

Unlike apoptosis or necrosis, ferroptosis is driven by the accumulation of reactive oxygen species (ROS) and the subsequent damage to cellular membranes. The discovery of ferroptosis has opened new avenues in cancer research, particularly in understanding how it can influence anti-tumor immunity. Ferroptosis is primarily initiated by the depletion of glutathione (GSH) or the inhibition of glutathione peroxidase 4 (GPX4). This depletion results in an inability to neutralize lipid peroxides, leading to oxidative damage and cell death. Iron, a critical component in the formation of ROS through the Fenton reaction, plays a central role in this process. Cancer cells often exhibit altered iron metabolism, rendering them particularly susceptible to ferroptosis. The induction of ferroptosis in cancer cells has shown promise as a therapeutic strategy, especially in tumors that are resistant to conventional therapies [92,93]. A large number of therapeutic strategies based on inducing ferroptosis in tumor cells and ferroptosis agonists are being gradually developed. Recent studies have indicated that, in addition to tumor cells, part of the immune cells may also undergo ferroptosis either spontaneously or under the influence of the tumor microenvironment. Ferroptosis in immune cells often adversely impacts their growth, development, and functional capabilities, thereby diminishing their anti-tumor immune responses [94,95].

Activated CD8+ T cells release IFN-γ, which enhances the susceptibility of tumor cells to ferroptosis [96]. The impact of ferroptosis occurring in T lymphocytes on themselves has also been gradually studied. One study demonstrated that fatty acid uptake, facilitated by the upregulation of CD36 expression in CD8+ T cells within the tumor microenvironment (TME), can trigger the onset of ferroptosis in these T cells, thereby impairing their anti-tumor capacity [97]. In T cells deficient in GPX4, the onset of ferroptosis enables these cells to mount defenses against virus- and pathogen-induced infections; however, it also inhibits their ability to proliferate. Notably, overexpression of GPX4 effectively mitigates T-cell dysfunction [98]. Similarly, ferroptosis also can exert pro-tumorigenic effects by participating in the growth and maturation of various immunosuppressive cells. Neutrophils, as key mediators of innate immunity, serve as the first line of defense against microbial invasion and infection. However, within the tumor microenvironment (TME), neutrophils are often pathologically activated by aberrant signals originating from tumors, transforming them into polymorphonuclear myeloid-derived suppressor cells (PMN-MDSCs) [99]. It has been demonstrated that lipid peroxidation, mediated by fatty acid transporter protein 2 (FATP2), supports the potent immunosuppressive effects of PMN-MDSCs in the TME, thereby promoting MDSC survival and inducing tumor progression in various mouse models [100]. Ferroptosis-induced generation of polymorphonuclear myeloid-derived suppressor cells (PMN-MDSCs) has been shown to negatively impact the immune response of effector T cells.

Similarly, B cells serve as the primary effectors and executors of humoral immunity. One study found that marginal zone B (MZ B) cells are distinct from B1 and follicular B2 cells, as MZ B cells engage in ‘trogocytosis’ to cannibalize dendritic cells. They utilize MHC molecules captured from these dendritic cells to communicate with CD4+ T cells in an antigen-presenting manner, thereby triggering an immune response. Furthermore, preventing MZ B cell ferroptosis using GPX4 has been shown to promote MZ B cell survival and enhance anti-tumor immunity [101]. A study has demonstrated that NK cells within the tumor microenvironment (TME) exhibit characteristics of ferroptosis and concurrent immune response dysfunction. Mechanistic investigations suggest that this phenomenon is associated with oxidative stress damage induced by ferroptosis and alterations in glucose metabolism [102,103]. The reversal of natural killer (NK) cell dysfunction and the enhancement of NK cell survival are achieved through the up-regulation of nuclear factor erythroid 2-related factor 2 (Nrf2) expression, in conjunction with the application of liproxstatin-1 (ferroptosis inhibitor) [104,105]. Another study has indicated that ferroptosis negatively impacts dendritic cell-mediated immune responses. The occurrence of ferroptosis in dendritic cells has been shown to impede their maturation. Likewise, tumor cells undergoing ferroptosis can induce dysfunction in dendritic cells, impeding their ability to activate specific T cells and suppressing adaptive immunity [106,107]. In terms of inflammatory response, sinensetin (SNS) exhibits both anti-inflammatory and antioxidant properties. In mouse models, SNS enhances the transcription levels of Nrf2 and its downstream target gene, superoxide dismutase 2 (SOD2), through the activation of the SIRT1 pathway. Concurrently, it suppresses the formation of the NLRP3 inflammasome, ultimately mitigating the inflammatory damage to mouse organs [108].

Finding suitable ferroptosis regulators to reduce ferroptosis in immune cells has important practical significance for enhancing tumor immunity.

#### 3.1.4. HDAC3 and the Nrf2/HO-1 Pathway in Ferroptosis

The Nrf2/HO-1 axis is an important inhibitory pathway for ferroptosis. Upon the onset of oxidative stress, Nrf2 undergoes nuclear translocation and binds to the antioxidant response element (ARE), leading to the elevated expression of downstream antioxidant genes, including heme oxygenase-1 (HO-1) and glutathione peroxidase 4 (GPX4). Increased levels of HO-1 protect cells from ferroptosis-induced oxidative stress by raising intracellular free iron levels and reducing reactive oxygen species (ROS) release [109]. The Nrf2/HO-1 pathway is essential for regulating intracellular iron metabolism homeostasis and mitigating the cytotoxic effects associated with excessive activation of the ferroptosis pathway. Likewise, the Nrf2 signaling axis is ubiquitous in immune cells and protects them from oxidative stress.

In an in vitro experiment involving melanoma cells, auranofin-pretreated NK cells demonstrated a reduction in the accumulation of intracellular reactive oxygen species (ROS) in an Nrf2-dependent manner. This treatment mitigated oxidative stress-induced damage and restored the immune killing function of NK cells [105]. Epstein–Barr virus (EBV) induces oncogenic transformation of B lymphocytes by inhibiting mitophagy in B lymphocytes to promote the release of reactive oxygen species (ROS). The p65 protein reduces the increase in ROS levels by interacting with NRF2, thereby preventing the malignant proliferation of B lymphocytes [110]. Asparagine restriction activates the NRF2-dependent stress response and increases the nucleotide content in CD8+ T cells, thereby promoting their proliferation and enhancing the efficacy of the immune response [111]. In another study, Baicalein inhibits MDSC proliferation and inflammatory response by inhibiting the activation of NLRP3 inflammasome in MDSC and upregulating Nrf2/HO-1 signaling, ultimately alleviating renal function damage in lupus mice [112]. In a glioblastoma-based rat model, the inhibition of glioblastoma-derived exosome (GDE) secretion effectively suppressed glioblastoma growth. Mechanistically, GDEs downregulated the expression of NRF and GPX4, which exacerbated lipid peroxidation levels in dendritic cells (DCs), induced ferroptosis in dendritic cells, and led to immune dysfunction. Knocking down the expression of Rab27a inhibits the secretion of GDE, reduces the level of ferroptosis in DC, and ultimately restores the vitality of DC [113].

There is sufficient evidence that histone deacetylase 3 (HDAC3) significantly regulates the Nrf2/HO-1 pathway. In a surgical brain injury (SBI) rat model, the administration of RGFP966 (an HDAC3 inhibitor) resulted in upregulation of NRF2 and HO-1 levels, as well as superoxide dismutase 2 (SOD2) expression, thereby reducing SBI-mediated oxidative stress damage to neuronal cells [114]. In the OVE26 diabetic mouse model, the administration of RGFP-966 (an HDAC3 inhibitor) facilitated the synthesis of Fibroblast Growth Factor 21 (FGF21) in the liver and decreased the incidence of diabetes-related aortic lesions. This effect aligns with the silencing of HDAC3, which enhances miR-200a expression, thereby upregulating Nrf2 transcription levels and reducing Keap1 expression [115]. Inhibiting HDAC3 enhanced the permeability of the blood–brain barrier (BBB) in spinal cord injured (SCI) rats by activating the SIRT1-Nrf2 axis and up-regulating the transcription levels of downstream targets, including HO-1 and quinone oxidoreductase (NQO1). This intervention also reduced inflammation associated with spinal cord injury. Based on the evidence presented, we can speculate that the classic antioxidant pathway Nrf2/HO-1 is present in most immune cells. HDAC3 inhibitors may serve as promising ferroptosis inhibitors by activating the Nrf2/HO-1 axis, which inhibits the ferroptosis of immune cells and restores their effector functions. This activation has the potential to awaken and enhance anti-tumor immunity, thereby providing clinical benefits to patients [116].

#### 3.1.5. HDAC3 in Regulating Retinoic Acid (RA) Signals for Immune Regulation

Retinoic acid (RA) is the primary bioactive metabolite of retinol (vitamin A). It plays a significant regulatory role in both innate and adaptive immunity by influencing the growth, differentiation, and migration of immune cells. The immune effects mediated by RA depend on the specific cell type and the site of RA release within the immune microenvironment. In the body, retinoic acid exists in various forms, such as 9-cis and all-trans, and interacts with retinoic acid receptors (RAR) to regulate the transcription of downstream target genes [117]. Adapalene, an agonist of the retinoic acid receptor (RAR), effectively promotes the cellular senescence-associated secretory phenotype (SASP) and enhances the tumor clearance of prostate cancer by natural killer (NK) cells [118]. Evidence suggests that retinoic acid (RA) plays a critical role in the immune regulation of the organism. For instance, RA promotes the development and maturation of fetal lymphoid tissue inducer (LTi) cells during pregnancy, thereby enhancing the offspring’s immune response. Additionally, RA regulates the balance between Th17 and Treg cells in accordance with the conditions and status of the immune microenvironment, thereby maintaining immune homeostasis [119,120].

Retinoic acid (RA) is also an important cofactor for the activation and enhancement of humoral immunity, effectively promotes the renewal and maturation of B lymphocytes, and induces their differentiation into plasma cells. RA significantly up-regulates the expression of B lymphocyte-induced maturation protein 1 (Blimp-1) in germinal center (GC) B lymphocytes and activates the expression of activation-induced cytidine deaminase (AID) in these cells, thereby supporting the normal release of immunoglobulin G (IgG) [121,122,123]. Furthermore, RA signaling activates follicular dendritic cells (FDCs) via retinoic acid receptors (RAR) and up-regulates the expression of chemokine ligand 13 (CXCL13) and B cell activation factor (BAFF), processes that facilitate B cell survival and migration in the gut and enhance the immune response [124,125,126].

Similarly, the percentage of NK cells in the peripheral circulatory system is positively correlated with the level of RA, which upregulates the expression of MHC class I chain-related molecules A (MICA) in tumor cells. MICA initiates the NK cell immune response by binding to the natural killer cell group 2D (NKG2D) receptor on the surface of NK cells [127].

The relationship between intestinal mucosal immune dysregulation and the development of intestinal malignant tumors has garnered increasing attention and research. The excessive release of a variety of pro-inflammatory factors will induce the occurrence of inflammatory bowel disease, seriously damaging the integrity of the intestinal barrier and the homeostasis of intestinal mucosal immunity. All-trans retinoic acid (atRA) inhibited the release of inflammatory mediators, including TNF-α, NO, IL-12, PGE2, and COX-2, in lipopolysaccharide (LPS)-activated macrophages, thereby inhibiting the occurrence of excessive inflammatory responses [128]. RA induces the differentiation of DC precursors (pre-DC) to CD103+CD11b+DC via the intestinal transit receptor α4β7, and this subtype of DC mediates the generation of Foxp3+ regulatory T cells and IL-10-producing effector T cells to alleviate pathogen-induced intestinal inflammation. In addition, CD103+CD11b+ DC synthesis releases retinoic acid into the peripheral circulatory system, which mediates intestinal homing of immune effector cells by inducing target cells to synthesize and express α4β7 and chemokine receptor 9 (CCR9) on the cell membrane surface, and ultimately through the α4β7/MAdCAM-1 and CCR9/CCL25 axis. During the inflammatory response phase, RA induces DCs to produce inflammatory mediators and promotes the differentiation of effector T cells while inducing the formation of tertiary lymphoid structures (TLS) and driving an adaptive immune response. These physiological processes mediated by retinoic acid are essential for the maintenance of intestinal immune homeostasis [129,130,131,132].

There is still more evidence that retinoic acid plays a non-negligible role in activating and maintaining immune responses [133]. Existing evidence shows that HDAC3 achieves immune regulation by affecting the transformation of the RA signal. HDAC3 negatively regulates the synthesis and function of retinoic acid, while the transcription factor JDP2 inhibits retinoic acid (RA)-dependent transcription by recruiting the histone deacetylase 3 (HDAC3) complex to the promoter regions of target genes. Additionally, JDP2 inhibits the activation of the p300/ATF-2 axis through the recruitment of HDAC3. This recruitment results in a decrease in RA-induced c-jun gene transcription and cell differentiation [134,135,136]. Additionally, butyric acid, a short-chain fatty acid (SCFA), enhances RA expression in both human and mouse epithelial cells through the inhibition of HDAC3. Similarly, the application of HDAC3 inhibitors has been shown to produce comparable effects [137]. Ski protein is a negative regulator of retinoic acid (RA) signaling. Ski inhibits retinoic acid (RA) signaling by interacting with key components of HDAC3 and acting as a transcriptional corepressor. Therefore, silencing HDAC3 or using HDAC3 inhibitors may facilitate the synthesis and normal function of retinoic acid, thereby exerting beneficial effects on maintaining and enhancing anti-tumor immune responses [138,139] (Figure 2).

#### 3.1.6. HDAC3 and CXCL8/CXCR2 Signaling Axis

The chemokine CXCL8, also known as interleukin-8 (IL-8), is a pro-inflammatory chemokine that plays a crucial role in recruiting neutrophils to sites of inflammation, infection, or injury. In the context of cancer, CXCL8 is produced by various cell types within the tumor microenvironment (TME), including infiltrating immune cells, stromal cells, and tumor cells [140]. Recent studies have demonstrated a relationship between CXCL8 and components of the TME, revealing new crosstalk mechanisms that can promote tumor progression and potentially establish a positive feedback loop. Immune checkpoint inhibition (ICI) has emerged as a cornerstone of immunotherapy for various cancers, and recent trials underscore the critical role of CXCL8 in the efficacy of ICI treatments [141,142].

CXCR2 is a primary receptor for CXCL8 and is classified within the CXC chemokine receptor family. High expression levels of CXCR2 in tumor cells are often associated with malignant progression, angiogenesis, metastasis, and immune evasion in tumors. Inhibition of CXCR1/2 signaling has been shown to synergize with PD-1/PD-L1 treatment, resulting in a reduction of mesenchymal tumor characteristics in mouse models of breast and lung cancer. This combined approach significantly increases the expression of epithelial E-cadherin while concurrently reducing the infiltration of granulocyte myeloid suppressor cells. Consequently, this enhances T cell infiltration and activation within tumors, thereby improving anti-tumor activity. Furthermore, there is a potential benefit in the combined blockade of CXCR1/2 and TGF-β signaling, which may modulate tumor plasticity and enhance tumor response to the PD-L1 blockade [143]. Research indicates that the combination of CXCR2 antagonists and anti-PD1 therapy can reduce tumor burden and extend patient survival. This combination treatment leads to increased activation of intratumoral XCR1-type dendritic cells and an elevation in CD8 T cell numbers, with evidence suggesting that genetic impairment of myeloid cell recruitment, neutralization of the XCR1 ligand XCL1, or depletion of CD8 T cells can diminish therapeutic efficacy [144].

The CXCL8/CXCR2 axis plays a significant role in tumor microenvironment (TME) remodeling, cancer plasticity, and the development of resistance to both chemotherapy and immunotherapy. Some studies have indicated the potential of CXCL8 as a biomarker for predicting resistance to immune checkpoint inhibitors (ICIs). Patients exhibiting higher baseline levels of CXCL8 tend to have a poor clinical prognosis. Moreover, clinical data suggest that CXCL8 may serve as an easily measurable prognostic biomarker in patients undergoing treatment with immune checkpoint inhibitors [145]. Evidence indicates that elevated baseline levels of CXCL8 in plasma correlate with worse clinical outcomes in patients receiving immunotherapy, such as nivolumab or ipilimumab, implying that CXCL8 could be utilized as a biomarker to predict the efficacy of ICI treatment [146]. Additionally, CXCR2 (CXC chemokine receptor type 2), which is often highly expressed in patients with hepatocellular carcinoma, is also recognized as a poor prognostic marker across various tumor types, including liver cancer [147].

The mutual binding of CXCL8 to CXCR2 activates downstream signaling pathways and enhances the invasiveness of various tumors, including hepatocellular carcinoma (HCC), making it a potential therapeutic target. In the context of immunotherapy, studies have demonstrated that elevated levels of CXCL8 correlate with a poor response to immune checkpoint inhibitors (ICIs). Recent research indicates that the CXCL8/CXCR2 signaling pathway is crucial for the recruitment of myeloid-derived suppressor cells (MDSCs) into the tumor microenvironment (TME), with MDSC accumulation contributing to immune suppression and tumor progression. Furthermore, studies have shown that combined treatment with CXCL8/CXCR2 inhibitors, such as SX-682, and ICIs can significantly enhance anti-tumor efficacy by reducing MDSC recruitment [148,149].

The above evidence shows that the activation of the CXCL8/CXCR2 axis inhibits anti-tumor immunity while promoting the invasiveness and immune escape of tumor cells. Consequently, CXCL8/CXCR2 serves not only as a potential biomarker for the prognostic evaluation of HCC patients but also presents significant potential as a therapeutic target.

HDAC3 has been shown to regulate the expression of CXCL8. Inhibition of HDAC3 can disrupt the CXCL8/CXCR2 axis, which reduces the infiltration of granulocytic myeloid-derived suppressor cells (G-MDSC) and enhances the activation and infiltration of T cells [150,151]. This alteration improves the tumor immune microenvironment and facilitates the transformation of “cold tumors” into “hot tumors”, thereby enhancing the efficacy of PD-1/PD-L1 checkpoint inhibitors [152]. Consequently, the combination of HDAC3 inhibitors (HDAC3i) and tumor immunotherapy is regarded as a promising new treatment strategy with significant clinical applications and research potential, offering new opportunities to improve the overall prognosis of cancer patients (Figure 3).

### 3.2. HDAC3 Regulating Immune Cells by Different Mechanisms

Existing evidence demonstrates that the regulation of immune cells by HDAC3 is frequently bidirectional. On one hand, HDAC3 is a crucial factor in the growth, development, and maturation of immune cells. For instance, HDAC3 serves as a crucial regulator during T cell development and maturation. T cells that lack HDAC3 are unable to undergo positive selection, resulting in the failure to generate CD4+ or CD8+ single-positive cells from the CD4-CD8-double-negative stage. This process is vital for maintaining peripheral T cell homeostasis [153].

On the other hand, HDAC3 can induce immune cell dysfunction through multiple signaling pathways, thereby impairing the anti-tumor immune response.

#### 3.2.1. HDAC3 and T Lymphocytes

Mature T cells are distributed to peripheral immune organs through the bloodstream, where they perform functions such as cellular immunity and immunomodulation. They are efficiently activated and recruited to infiltrate the tumor microenvironment (TME), thereby exerting anti-tumor immune efficacy through various mechanisms. This process is a critical component of many current immunotherapies in clinical use.

But, on the other hand, there is evidence that inhibiting HDAC3 expression will help restore or enhance the anti-tumor immune function of CD8+ T cells. For instance, the interaction between the 4-1BBL/CD157 and OX-40L/CD252 ligands serves as a positive activator of effector T cells. In a chemotherapy-resistant ovarian cancer A2780-AD cell model, the use of HDAC3 inhibitors significantly increased the expression of OX-40L and 4-1BBL in the cells [154]. Thymic medullary epithelial cells (mTEC) are essential for the generation of Foxp3(+) regulatory T cells, and their normal function is highly dependent on HDAC3. Interleukin-2 (IL2) is a cytokine crucial for the polarization of naive T cells into mature T cells. Inhibition of HDAC3 has been shown to upregulate IL2 promoter activity and impede the differentiation of naive T cells into the FOXP3+ Treg subset. This process induces the release of significant amounts of IL-2, IL-6, and IL-17 while simultaneously blocking the immunosuppressive function of Tregs (regulatory T cells) [155]. The transcriptional regulator oxidative stress response protein 2 (Osr2) is significantly enriched in tumor-infiltrating PD-1+ and TIM-3+ end-stage T cells. In the MC38-OVA colon cancer cell model, the recruitment of HDAC3 protein through the Piezo1/calcium/CREB axis inhibits the expression of cytotoxic genes, thereby contributing to the development of an exhausted CD8+ T cell phenotype. The application of HDAC3 inhibitors may aid in rescuing exhausted T cells and reducing immune evasion [156]. A novel selective HDAC3 inhibitor, HQ-30, has been demonstrated to enhance anti-tumor immune responses by regulating PD-L1 expression. HQ-30 targets cathepsin B (CTSB) to mediate PD-L1 degradation, which leads to increased infiltration of CD4+ T cells and CD8+ T cells in tumors, thereby reshaping the tumor microenvironment [157].

#### 3.2.2. HDAC3 and B Lymphocytes

B lymphocytes comprise 8–15% of peripheral blood lymphocytes. Upon antigen stimulation, they proliferate and differentiate into mature plasma cells, which then rapidly secrete large amounts of antibodies represented by IgG to stimulate the immune response. B lymphocytes play a crucial role in humoral immunity. But, at the same time, the malignant proliferation and differentiation of B cells and the initiation of incorrect epigenetic programs can lead to the occurrence of malignant diseases. Germinal center (GC) B cells are responsible for many B-cell malignancies. Epigenetic dysregulation, resulting from mutations in two genes, KMT2D and CREBBP, in germinal center (GC) B cells, accelerates the development of B-cell non-Hodgkin lymphomas (B-NHLs). Selective HDAC3 inhibitors restore the aberrant epigenetic programming of CREBBP by directly targeting the BCL6-HDAC3 repressor complex [158]. Genetically engineered mice with Cd19-Cre-Hdac3−/− exhibit impaired germinal center formation and suppressed plasma cell production, a result that correlates with the HDAC3 deletion. This result is related to the changes in BCL6, SMRT, and FOXO1 transcript levels affected by HDAC3 deletion and regulates the formation of germinal center-derived cells and reduces B-cell malignancy through its direct targets CXCR5, CD86, and CD83 [159]. In a cellular model of diffuse large B-cell lymphoma (DLBCL), the inhibition of HDAC3 rescues the aberrant epigenetic programming associated with CREBBP mutations, enhancing BCL6-mediated antigen presentation and the release of IFN-γ. This intervention restores the ability of tumor-infiltrating lymphocytes to effectively kill cancer cells in an MHC class I/II-dependent manner. Furthermore, HDAC3 inhibition exhibits synergistic effects when combined with PD-L1 inhibitors [160,161]. These findings reveal the uncommon potential of HDAC3 inhibitors in the treatment of B-cell lymphomas.

#### 3.2.3. HDAC3 and Natural Killer (NK) Cells

Natural killer (NK) cells are recognized for their autonomous and non-specific ability to kill target cells, independent of antigenic stimulation. Within the context of innate immunity, they serve as the primary effector cells involved in the body’s anti-tumor immunity, responses to viral infections, and immunomodulation. The regulatory role of HDAC3 on NK cells has been gradually studied. In mouse models of peripheral T-cell lymphoma (PTCL), drugs that specifically inhibit HDAC3 are employed to enhance the efficacy of anti-tumor therapies. The inhibition of HDAC3 can elevate the secretion of CXCL12 cytokines through the regulation of ATF3 signaling, recruit and activate natural killer (NK) cells, and improve the sensitization effects of radiotherapy [162]. ULBP proteins function as NKG2D ligands, facilitating antigen clearance by activating NK cell cytotoxicity. However, HDAC3 mediates immune evasion in tumor cells by interacting with the transcription factor SP3, which inhibits ULBP1 protein expression. Trichostatin A, an HDAC3 inhibitor, has been shown to effectively upregulate the expression of ULBP ligands in tumor cells, thereby activating NK cell-mediated anti-tumor immunity [163,164]. Similarly, in a small cell lung cancer (SCLC) model, the proto-oncogene c-Myc recruits HDAC3 to the promoter region of the MICA/B gene, negatively regulating the expression of the NK cell activating receptor (NKG2DL) through histone H3K9 deacetylation, which inhibits NK cytotoxicity and mediates immune inhibition [165].

#### 3.2.4. HDAC3 and Dendritic Cells (DCs)

Dendritic cells are regarded as the most potent and specialized antigen-presenting cells (APCs), capable of efficiently uptaking, processing, and presenting antigens while simultaneously activating primary T cells. They act as central messengers in the initiation, regulation, communication, and sustenance of immune responses. Type I interferon (IFN-I) is an important activator of dendritic cells. In Namalwa lymphoma cells, HDAC3 promotes the acetylation of histone H3K9/K14, which hinders the binding of the interferon regulatory factor IRF7 and the TATA-binding protein (TBP) in the promoter region, thereby inhibiting the expression of the IFN-γ factor [166]. Moreover, the vitamin D receptor (VDR) suppresses relB gene transcription and dendritic cell maturation by recruiting complexes that include HDAC3 [167]. In an organoid-based study of breast cancer, the use of adenovirus to block the interaction of HDAC3 with nuclear receptor co-repressor 2 (NCOR2) was found to up-regulate the expression of interferon regulatory factor 1 (IRF-1) and interferon-γ (IFN-γ) factors. This release of IFN-γ promoted the activation of dendritic cells and enhanced anti-tumor immunity, which improved drug resistance in patients and increased the efficacy of immune checkpoint inhibitors [15]. The above evidence indicates that inhibiting HDAC3 helps to successfully activate DC cells and induce their maturation to initiate downstream anti-tumor specific immune responses.

#### 3.2.5. HDAC3 and Macrophages

Macrophages possess the capability to efficiently ingest, engulf, and degrade pathogens. They can be categorized into two primary phenotypes: M1 and M2. M1 macrophages mediate the immune response to eliminate foreign microorganisms by secreting pro-inflammatory factors; however, they are also associated with tissue damage and persistent inflammation. In contrast, M2 macrophages are pivotal in responses to parasites, angiogenesis, and allergic diseases, but they play a significant immunosuppressive role in anti-tumor effects. It has gradually become a consensus that inflammation promotes the occurrence and progression of cancer. Sustained inflammation has become a high-risk factor for inducing tumors and has a negative impact on the prognosis of patients undergoing surgical treatment. The tumor microenvironment facilitates the advancement of malignant tumors and mediates immune suppression by recruiting and activating inflammatory cells [168]. M1 macrophages play a key role in the development of many chronic inflammatory and autoimmune diseases [169]. HDAC3 can trigger chronic inflammation in the IL-1β-dependent pathway by regulating NLRP3/caspase−1 signaling [170]. The use of RGFP966 (HDAC3 inhibitor) has been demonstrated to effectively modulate the inflammatory signaling pathway in macrophages, resulting in a reduction of IL-6 and TNF-α release [171,172]. Additionally, BG45, another HDAC3 inhibitor, has been demonstrated to significantly reduce the expression of M1 macrophage markers in tissues and promote the polarization of macrophages toward the M2 phenotype, thereby alleviating the impact of severe inflammation on the body. While inducing the polarization of macrophages toward the M1 phenotype generally enhances anti-tumor immune responses, the role of inflammation in promoting cancer should not be overlooked [173]. Inhibiting HDAC3 may mitigate the damage and risks associated with the excessive inflammatory response induced by M1 macrophages. This approach has potential implications for enhancing the efficacy of immunotherapy and maintaining immune homeostasis in vivo (Figure 4).

## 4. HDAC3 Acting as a Therapeutic Target in Cancer Immunotherapy

### 4.1. HDAC3 Inhibition in Cancer Immunotherapy

Based on the evidence presented previously, it is evident that HDAC3 plays a significant role in regulating immune cells and signaling pathways. The use of HDAC3 inhibitors can enhance the effector functions of natural killer (NK) cells, CD8+ T cells, and dendritic cells, while also mitigating immune suppression, such as that caused by myeloid-derived suppressor cells (MDSC). Inhibition of HDAC3 may prevent ferroptosis-induced immune cell dysfunction by activating the NRF2/HO-1 pathway. Additionally, activating the retinoic acid (RA) pathway can enhance effector cell-mediated anti-tumor immune responses. The transcriptional expression of HDAC3 plays a significant role in the activation of the CGAS-STING signaling pathway. Overexpression of HDAC3 may facilitate tumor cell apoptosis and promote the release of factors such as IL-2 and IFN-γ by activating the CGAS-STING pathway. These factors can enhance anti-tumor immunity, ultimately providing clinical benefits to patients.

Therefore, the use of HDAC3 inhibitors can effectively enhance anti-tumor immune responses. Further therapeutic strategies based on HDAC3 inhibitors warrant exploration and development.

### 4.2. Inhibition of HDAC3 for Improving Anti-Cancer Immunity

Tumor immunotherapy differs from conventional therapies, including surgery, radiotherapy, and chemotherapy. Its primary objective is to enhance and reactivate the host’s anti-tumor immune response by increasing the immunogenicity of tumor cells and stimulating the autoimmune system, and this approach has clinical advantages such as durability, thoroughness, and comprehensiveness. Although the drug response rate remains limited, immunotherapy continues to provide clinical benefits for the majority of cancer patients. The ecology of tumor immunity is inherently complex. Under normal circumstances, the immune system recognizes and eliminates tumor cells within the host. However, tumor cells have evolved various strategies and mechanisms to evade this immune response, including the recruitment of immunosuppressive cells, the impairment of antigen presentation, and the induction of effector cell dysfunction. This result contributed to tumor survival during various stages of anti-tumor immunotherapy [174].

#### 4.2.1. Existing Antitumor Immunotherapy

Understanding the underlying mechanisms of interactions between targets and cells within the tumor microenvironment is crucial for the development of new immunotherapies and the enhancement of treatment response rates. Currently, tumor immunotherapy, grounded in progressively mature mechanistic theories and technologies, has achieved significant advancements in both basic and clinical research. Tumor immunotherapy, represented by oncolytic virus therapy, tumor vaccines, adoptive cell transfer therapy, immune checkpoint inhibitors, and cytokine therapy, has become a hot topic in the field of cancer treatment [175,176]. Oncolytic virus therapy represents one of the earliest forms of immunotherapy, using modified viruses to target cancer cells. When these viruses infect tumors, they trigger inflammation and activate the body’s immune defenses against cancer. T-Vec, a modified herpes simplex virus, has proven particularly effective against advanced melanoma that cannot be surgically removed [177]. Cancer vaccines work by introducing inactive tumor antigens to stimulate T cell responses against cancer. The melanoma antigen GP100, for example, has successfully activated tumor-fighting immune cells. In prostate cancer treatment, Provenge stands out as the first FDA-approved immunotherapy vaccine. This treatment uses the patient’s own immune cells combined with the PA2024 antigen to fight metastatic castration-resistant prostate cancer [178,179]. By collecting the patient’s peripheral blood mononuclear cells and co-culturing them with the antigen PA2024 in vitro and subsequently infusing these cells back into the patient, dendritic cells (DCs) can be effectively induced to mature and activate T cell-mediated immune responses that identify and eliminate prostate cancer cells expressing the PAP antigen. These findings underscore the unique role of cancer vaccines in cancer treatment [180]. Immune checkpoint inhibitors (ICI) have revolutionized cancer treatment by targeting natural immune system regulators. These checkpoints normally prevent autoimmune responses, but cancer cells exploit them to hide from immune attack. Two key checkpoints, PD-1 and CTLA-4, act like brakes on the immune system. CTLA-4, found on regulatory T cells, suppresses immune responses by blocking the stimulatory signals that normally occur when B7-1/2 molecules bind to CD28 receptors on T cells [181]. Several immune checkpoint inhibitors have proven highly effective in cancer treatment. These include ipilimumab, which targets CTLA-4; nivolumab and pembrolizumab, which block PD-1; and avelumab, durvalumab, and atezolizumab, which target PD-L1. Ipilimumab, the first approved checkpoint inhibitor, works by preventing CTLA-4 from binding to its receptor, allowing T cells to mount a sustained immune response. Similarly, nivolumab disrupts the interaction between PD-1 and PD-L1, preventing tumors from evading immune detection and restoring T cells’ ability to attack cancer cells. Research continues to uncover new immune checkpoints, including TIGIT, B7-H6, TIM-3, LAG-3, BTLA, and IDO2. Scientists are actively studying these targets to develop the next generation of checkpoint inhibitors [182]. Adoptive cell transfer therapy harnesses the body’s own immune cells to fight cancer. The process begins by collecting immune cells—either from the patient’s blood, tumor tissue, or from donors. These cells are then activated and multiplied in the laboratory using compounds like IL-2 and anti-CD3 antibodies before being returned to the patient. Two advanced forms of this approach have shown remarkable promise. CAR-T therapy engineers T cells to carry specialized receptors that recognize specific tumor markers, enabling them to release powerful immune signals and precisely target cancer cells. TCR-T therapy represents another breakthrough in this field, adding to the growing arsenal of cellular immunotherapies. These techniques have proven particularly effective in cancer treatment, marking a significant advance in personalized immunotherapy [183,184]. TCR engineering of tumor-infiltrating lymphocytes (TILs) represents one of the most promising treatments for tumors, achieved by transducing chimeric antigen receptors or TCRα/β heterodimers into TILs to enhance their specific binding affinity to tumor-associated antigens (TAAs). Kimmtrak (tebentafusp-tebn), the first bispecific TCR-T therapy approved by the FDA, has demonstrated breakthrough results in the treatment of unresectable or metastatic uveal melanoma (mUM) [185]. Similarly, satisfactory anti-tumor efficacy was clinically observed after TCR genes were transferred to the peripheral blood lymphocytes (PBLs) of melanoma patients [186]. Cytokine therapy fights cancer by using immune-stimulating proteins like IL-2 and IFN-α to activate and multiply immune cells. These cytokines act as cellular messengers, promoting dendritic cell maturation and enhancing T cells’ ability to kill cancer cells. To minimize side effects and improve patient tolerance, cytokines are often used alongside cell transplant therapies [187]. Immunotherapy has become a cornerstone of modern cancer treatment, with T cells playing the central role. The approach focuses on four key strategies: triggering cytokine release, activating T cells’ cancer-fighting abilities, preventing T cell exhaustion, and introducing specially engineered T cells that can recognize and target cancer cells.

#### 4.2.2. Challenges and Obstacles of Antitumor Immunotherapy

Although anti-tumor immunotherapy has made significant progress in recent years, several deficiencies and challenges continue to hinder its application in clinical practice. A primary challenge is the low response rate; for instance, anti-PD-1/PD-L1 therapy is effective only for a subset of patients, with response rates typically ranging from 20% to 40%. Consequently, many patients do not derive any benefit from this treatment. This phenomenon is attributed not only to significant individual differences among patients but also to the high heterogeneity observed in certain tumor cells, which can render immunotherapy ineffective for specific tumor subtypes. Additionally, some tumors exhibit low immunogenicity, commonly referred to as “cold tumors”, and the physical isolation provided by the extracellular matrix can impede the infiltration of immune cells. In tumor microenvironments, there exists a significant presence of immunosuppressive cells, including Tregs and MDSCs, along with elevated levels of immunosuppressive cytokines and metabolic products. This collaborative effect weakens the immune response, making it challenging for the immune system to identify and attack tumor cells. Under certain conditions, this tumor immunosuppressive microenvironment can even induce primary or acquired drug resistance in patients undergoing immunotherapy. These factors complicate the exploration and development of effective immunotherapy strategies.

#### 4.2.3. HDAC3 Inhibitors Have the Potential to Become Sensitizers of Immunotherapy

Tumor immune escape is a major challenge currently faced in tumor immunotherapy. Tumor cells can secrete immunosuppressive factors and induce the generation of immunosuppressive cells, or they may rely on ligand pathways such as FAS/FAS-L and PD1/PDL1 to activate and amplify immunosuppressive signals; this enables them to evade recognition and attack by the host’s immune cells [188]. Research on HDAC3 inhibitors as sensitizers for tumor treatment has been reported. In an in vitro experiment, the knockout of HDAC3 using CRISPR/Cas9 resulted in increased DNA damage in irradiated tumor cells, thereby reversing the resistance of rhabdomyosarcoma (RMS) to radiotherapy and exerting a radiosensitization effect [2]. Additionally, one study demonstrated that the use of HDAC3 inhibitors promoted AKT protein acetylation, which enhanced the sensitivity of leukemia cells to chemotherapy drugs [189]. It is exciting that the current research shows that the HDAC inhibitor has gradually shown good potential in the effect of sensitizing tumor immunotherapy.

In gastric cancer cell models, the knockdown of HDAC3 inhibited the nuclear translocation of STAT1, impaired IFN-γ signaling, and reduced B7-H1 expression in HGC-27 cells. Ultimately, the percentage of infiltrating CD8+ T cells in the tumor microenvironment is increased, and the immune evasion of tumor cells is inhibited [190]. In addition, in the B16 melanoma cell model, treatment with AR42 (a histone deacetylase inhibitor) resulted in a reduction of PD-L1 and PD-L2 expression in B16 cells, while simultaneously increasing the infiltration of neutrophils and natural killer (NK) cells. Within the tumor microenvironment (TME), it enhances the body’s immune response to tumor cells [45]. Research using a mouse fibrosarcoma model shows that HDAC3 suppresses the CXCL9/10/11-CXCR3 pathway by binding to chemokine promoters. When HDAC3 is blocked, it slows the growth of MCA205 tumors and draws more CXCR3+ T-cells into the tumor environment. This leads to increased numbers of CD4+IFNγ+, CD8+IFNγ+, and CD11b+F4/80+ immune cells in the tumors, ultimately helping to destroy cancer cells [191]. Therefore, the above evidence indicates that targeting HDAC3 is an effective strategy that may enhance anti-tumor immunity.

The above studies have indicated that inhibiting the expression of HDAC3 can significantly enhance the functionality of immune cells, particularly T cells. This inhibition promotes T cell-mediated anti-tumor immune responses by upregulating costimulatory signals such as OX-40L and 4-1BBL, facilitating the release of IL-2, and reducing the expression of immune checkpoints. Additionally, this inhibition impacts other immune cells, including NK and dendritic cells (DC), exerting a positive stimulatory effect. Studies have shown that class I HDAC inhibitors (HDACi), specifically chidamide, enhance the persistence and anti-tumor effects of CAR-T cells by upregulating the expression of transcription factors such as LEF1 and TCF4, as well as activating the Wnt/β-catenin pathway [192]. One study demonstrated nanomicelles that encapsulate siRNA-PD-L1 and combined them with HDAC inhibitors (HDACi) to effectively reverse T cell exhaustion and prevent the immune evasion of tumor cells. This discovery highlights the potential of epigenetic regulatory drugs and underscores the promising prospects of combining PD-1/PD-L1 blocking therapy [193]. Based on the evidence presented, while immunotherapy has achieved significant breakthroughs in clinical treatment, it still faces obstacles and challenges, including low response rates and unpredictable potential side effects. Nevertheless, both in terms of mechanisms of action and in light of existing research findings, there are compelling reasons to believe that HDAC3 inhibitors may serve as sensitizers when combined with immunotherapy, thereby enhancing the efficacy of immunotherapeutic approaches, underscoring the clinical value of tumor immunotherapy, and facilitating the exploration and development of novel clinical treatment strategies.

### 4.3. HDAC3 and Immune Checkpoint Inhibitor (ICI) Therapy

In the field of immunotherapy for tumors, research on immune checkpoint inhibitors (ICIs) has reached a mature and substantial stage. Numerous preclinical and clinical studies are continuously investigating combination therapies involving ICIs [194]. Notably, HDAC3 plays a dual role in regulating the expression of immune checkpoints on the cell surface. For instance, in a xenograft tumor mouse model constructed using A549/CDDP cells, overexpression of HDAC3 and inhibition of the JNK/c-Jun signaling pathway effectively suppressed PD-L1 expression and reduced chemoresistance in tumor cells [195]. Additionally, in a melanoma-based study, HDAC3 acted as a competitive receptor for DDX3X, forming an inhibitory complex with the retinoic acid receptor-related orphan receptor-alpha (RORA). This complex binds to the promoter region of CD274, inhibiting PD-L1 expression and thereby enhancing anti-tumor immune responses in T cells in vivo [196]. Immune checkpoint inhibitors targeting programmed cell death protein 1 (PD-1), programmed cell death-ligand 1 (PD-L1), cytotoxic T-lymphocyte-associated protein 4 (CTLA-4), lymphocyte activation gene-3 (LAG-3), and other immune checkpoints have demonstrated significant clinical efficacy in the treatment of various solid tumors. Recent studies indicate that low actual patient response rates are a significant obstacle preventing immune checkpoint inhibitors (ICIs) from achieving clinical efficacy. Research has been conducted to enhance the responsiveness of ICIs by increasing the abundance of specific gut flora. Taking PD-1 immune checkpoint inhibitors as an example, it is now a clinical consensus that higher PD-L1 expression rates correlate with greater efficacy of PD-1 immune checkpoint inhibitors. This conclusion is based on the clinical outcomes and efficacy of PD-1/PD-L1 immunotherapies, as well as the PD-1/PD-L1 expression levels in various cell types within the tumor microenvironment. Therefore, the clinical efficacy of PD-1/L1-based immunotherapy is closely linked to the PD-L1 expression levels in different cell types within the tumor microenvironment [197]. Therefore, understanding the biological mechanisms that regulate PD-L1 expression is crucial. In pancreatic ductal adenocarcinoma (PDAC) cell models, HDAC3 has been identified as critical for the regulation of PD-L1 expression. Further investigations revealed that HDAC3 mediates the activation of the Signal Transducer and Activator of Transcription 3 (STAT3) pathway, which in turn upregulates the transcriptional level of PD-L1 [35]. In a clinical study on thymic epithelial tumors (TET), PD-L1 expression was positively correlated with cytoplasmic expression of HDAC3 in TET tumor cells [198]. Additionally, in a mouse model of gastric cancer, it was observed that HDAC3 may induce B7-H1 expression by activating the JAK/STAT1 pathway [190]. In gastric cancer cells, activation of the SPI1-ZFP36L1-HDAC3 signaling axis promotes abnormal transcription of PD-L1 [199]. It is reasonable to propose that overexpressing HDAC3 may enhance the expression of various cell-surface immune checkpoints within the tumor microenvironment (TME). This finding could facilitate the combination of therapies targeting the epigenetic regulation of HDAC3 with immune checkpoint inhibitors, potentially benefiting patients with low expression levels of PD-1, PD-L1, and similar markers, thereby broadening the applicability of these treatments.

## 5. Cancer Prevention

Existing research indicates that the activation of the STING pathway may be associated with an increased risk of cancer in certain instances. While the STING pathway is crucial for enhancing anti-tumor immunity, its overactivation can result in sustained inflammatory responses, potentially leading to immunosuppression and pro-tumor effects. One study demonstrated that in the StingN153S/+ mouse model, the overactivation of the STING pathway disrupts calcium stability in T cells and mediates endoplasmic reticulum stress and T cell apoptosis via an IFN-γ-independent signaling. This study also showed that high doses of ADU-S100 (a STING agonist) mediate immune evasion by blocking the immune response of CD8+ T cells [200]. Another study demonstrated that the StingS365A/S365A mouse model reveals that B16 melanoma and MC38 colon cancer can induce immunosuppression by activating the STING pathway and promoting T cell death in an IFN-γ-independent manner [201]. In a pancreatic ductal adenocarcinoma mouse model utilizing the KPC4662 cell line, the activation of the STING pathway prompted regulatory B cells to express IL-35 in an IRF3-dependent manner [202]. This process subsequently reduced NK cell proliferation and impaired the NK cell-mediated anti-tumor immune effects. Therefore, regulating the activation homeostasis of the cGAS-STING pathway is crucial for achieving improved anti-tumor therapeutic outcomes.

The long-term inflammatory state and sustained activation of the STING pathway are thought to potentially promote the occurrence and progression of cancer. Chronic inflammation is closely associated with the pathogenesis of various cancer types and may increase cancer risk by promoting cell proliferation, inducing gene mutations, altering the microenvironment, or mediating the further progression of existing tumors [203,204,205]. Therefore, while activation of the STING pathway can contribute to anti-tumor immunity in certain contexts, its excessive or dysregulated activation may have detrimental effects and elevate the risk of tumor development. Researchers are investigating how to effectively modulate this pathway to develop therapeutic strategies that enhance immune responses without inciting excessive inflammation. The use of HDAC3 inhibitors may help mitigate excessive STING pathway activation as a potential preventive strategy against cancer development. However, further research is needed to explore relevant areas, such as the identification and study of biomarkers associated with excessive STING pathway activation and the correlation between the level of STING pathway activation and the onset of different tumors.

## 6. Future Direction and Challenges

The application of HDAC3 inhibitors as immunotherapy sensitizers in combination with immunotherapy presents a promising strategy for tumor treatment. However, it is important to note that the specific details regarding the combination of HDAC3 inhibitor and immunotherapy require further exploration and optimization.

### 6.1. Selecting Ideal Delivery Systems for HDAC3i

The blood viscosity environment within the host body is notably complex and variable. Numerous unpredictable factors, along with the host’s defense and clearance mechanisms against foreign invaders, significantly influence the efficacy of therapeutic agents. In addition, different nanomaterials exhibit targeting specificity towards various tumors, demonstrating significant effects in enhancing efficacy and reducing toxicity. Consequently, the selection of ideal drug delivery systems capable of effectively protecting and transporting HDAC3 inhibitors to target areas or cells is essential for effective therapy.

Nanodelivery systems represent a confluence of ongoing innovation and development in cancer treatment and nanomedicine [206,207]. The unique size and favorable biocompatibility of nanomaterials enhance safety and targetability in tumor treatment. Various nanodelivery systems have been progressively developed to meet diverse therapeutic expectations, including liposomes, mesoporous silica nanoparticles (MSNPs), gold nanoparticles (AuNPs), and pH-responsive nanomicelles [208]. These systems utilize personalized nanomaterials to encapsulate HDAC3 inhibitors, which not only protect HDAC3 from potential degradation threats but also control drug release. Most importantly, this protective drug-loading system can accurately deliver HDAC3 inhibitors to the targeted area and facilitate uptake by target cells. This intervention strategy can enhance the efficiency and precision of HDAC3 inhibitors treatment, effectively prolonging the action time of HDAC3 inhibitors while minimizing potential toxicity and side effects. In one study, cellulose was double-modified to create a nanocarrier that encapsulated HDAC3 siRNA, resulting in efficient uptake by HEK293 cells while preserving the integrity of the cell membrane as much as possible [209]. In addition, more potentially available drug delivery systems may be used to load HDAC3i to achieve targeted drug delivery, including integrated engineered exosomes, recombinant adeno-associated virus (rAAV), liposomes, and ADC (antibody-drug conjugate) system, PDC (peptide-drug conjugate) system, nano-nucleic acid carrier, etc., and more relevant clinical research is essential [210,211].

### 6.2. Challenges of Using HDAC3i in Combination with Immunotherapy

The combination of HDAC3 inhibitors (HDAC3i) with immunotherapy faces numerous challenges, including potential toxicity, off-target effects, and immune-related adverse events caused by immune system hyperactivation. In addition, variability in patient responses and the lack of reliable predictive biomarkers further complicate the optimization of combination strategies [212,213]. Specifically, the safety of these therapies is a critical issue, requiring in-depth research into the potential toxic side effects of HDAC3i at different doses and in combination regimens. Furthermore, exploring optimal dose–effect relationships, as well as the best dosing schedules and sequences, is essential to maximize therapeutic efficacy while minimizing toxicity risks. Future research should concentrate on elucidating the molecular mechanisms by which HDAC3 inhibitors (HDAC3i) affect tumor and immune cells. Additionally, it is important to explore and regulate biomarkers that may influence the efficacy of HDAC3i. Finally, well-designed preclinical and clinical studies should be conducted to evaluate its specific efficacy and safety. Integrated omics analysis and advanced modeling techniques can be employed to evaluate drug synergy and resistance mechanisms, paving the way for more personalized and effective combination therapies.

### 6.3. Exploration of Novel HDAC3i-Based Combined Therapeutic Strategies

In the quest to develop more effective and innovative cancer therapies, multi-modal treatment strategies have gained significant attention for their potential to overcome treatment resistance and improve patient outcomes [214]. The combination of HDAC3 inhibitors (HDAC3i) with immunotherapy, augmented by nano-delivery systems, and integrated with photodynamic therapy (PDT), photothermal therapy (PTT), and other treatment modalities, holds the potential to achieve triple-combination therapy. Additionally, pairing HDAC3i with two distinct immune checkpoint inhibitors or immunomodulatory agents may offer enhanced clinical efficacy. However, the feasibility, safety, and therapeutic effectiveness of these complex combinations demand rigorous evaluation through comprehensive basic and clinical research. Advancements in these areas are anticipated to significantly improve cancer remission rates and enhance the quality of life for patients.

Future research may focus on exploring and selecting multiple combination options. For instance, combining HDAC3 inhibitors (HDAC3i) with immunotherapy, utilizing a nano-delivery system for HDAC3i, and integrating photodynamic therapy (PDT), photothermal therapy (PTT), and other modalities could potentially achieve triple combination therapy. Additionally, combining HDAC3i with two different immune checkpoint inhibitors or immunomodulatory drugs may enhance clinical efficacy. However, the feasibility, safety, and effectiveness of these combination therapies require thorough evaluation through extensive clinical and basic research. Such advancements are expected to further improve the quality of life and remission rates for cancer patients.

## 7. Conclusions

HDAC3, as a pivotal epigenetic regulator, plays a multifaceted role in modulating the tumor microenvironment and immune responses, making it a promising target for immunotherapy sensitization. By influencing key pathways such as cGAS-STING, ferroptosis, and the Nrf2/HO-1 axis, HDAC3 inhibition has shown potential to reverse immune suppression, enhance antigen presentation, and improve the functionality of immune effector cells, including T cells, NK cells, and dendritic cells. These mechanisms underscore HDAC3’s capacity to transform “cold tumors” into “hot tumors”, thereby enhancing the efficacy of immune checkpoint inhibitors and other immunotherapeutic strategies. This work provides a comprehensive and novel perspective on HDAC3’s dual role in tumor progression and immunotherapy, highlighting its therapeutic potential both as a standalone target and as a critical component of combination therapies aimed at overcoming resistance and broadening the scope of immunotherapy. The holistic examination of HDAC3’s regulatory functions underscores the need for innovative approaches, including selective HDAC3 inhibitors and advanced drug delivery systems, to fully harness its clinical potential. While challenges such as toxicity, patient heterogeneity, and resistance mechanisms remain, refining combination strategies, optimizing dosing regimens, and developing predictive biomarkers will be essential to maximize the benefits of targeting HDAC3. As our understanding of its mechanisms deepens, HDAC3-based therapies are poised to play a transformative role in advancing tumor immunotherapy, ultimately improving outcomes for cancer patients worldwide.

## Figures and Tables

**Figure 1 vaccines-13-00182-f001:**
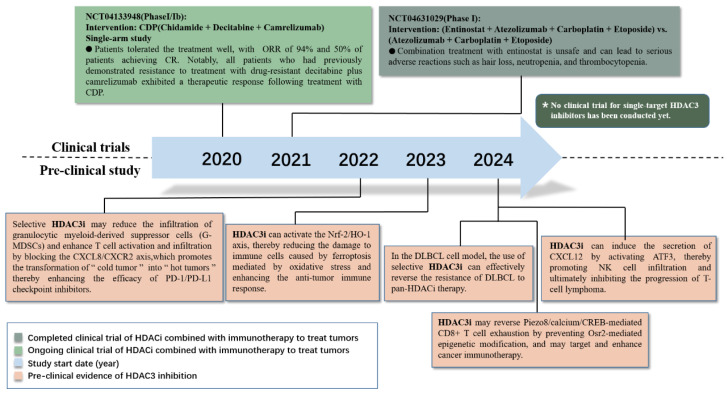
**Selected clinical trials of HDACi combined with immunotherapy and related preclinical study of HDAC3i in the last 5 years.** The use of pan-HADCi in combination with immunotherapy is feasible but may induce serious adverse reactions. Selective HDAC3 inhibitors enhance anti-tumor immune responses by preventing the recruitment of G-MDSCs, activating the Nrf-2/HO-1 antioxidant pathway, reversing CD8+ T cell exhaustion, and recruiting NK cells to the tumor microenvironment, thereby sensitizing the effects of PD-1/PD-L1 blockade therapy. Abbreviations: ORR, Objective Response Rate; CR, Complete Response; DLBCL, Diffuse large B cell lymphoma; CXCL12, C-X-C motif chemokine ligand 12; ATF3, Activating Transcription Factor 3; CREB, cAMP-response element binding protein; Osr2, Odd-Skipped Related Transcription Factor 2; Nrf-2, Nuclear factor erythroid 2-related factor 2; HO-1, Heme Oxygenase-1; CXCL8, C-X-C Motif Chemokine Ligand 8; CXCR2, C-X-C Motif Chemokine Receptor 2.

**Figure 2 vaccines-13-00182-f002:**
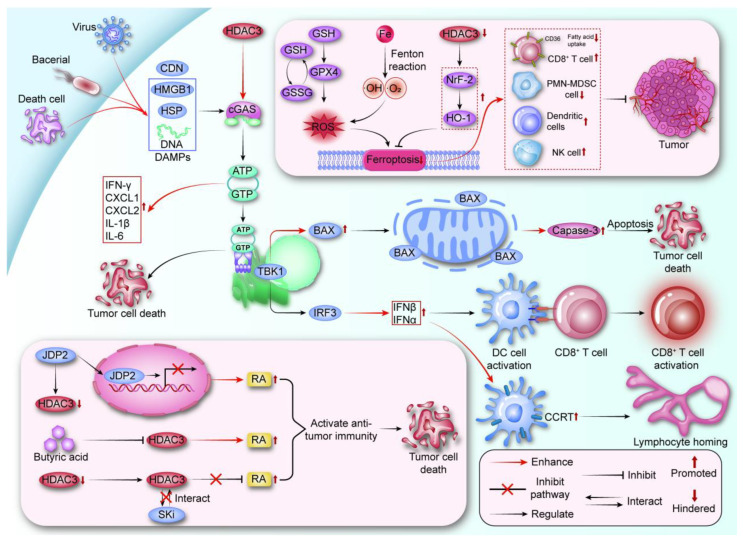
**Mechanisms of HDAC3 and HDAC3 inhibitors in modulating immune function and enhancing anti-tumor responses.** HDAC3i activates the Nrf-2/HO-1 axis by inhibiting HDAC3, therefore alleviating oxidative stress-mediated ferroptosis damage to immune cells and enhancing anti-tumor immune responses. 2. HDAC3 promotes the release of type I interferon by upregulating the transcriptional expression of cGAS, thereby enhancing the activation of CD8+ T cells, lymphocyte homing, and tumor cell apoptosis mediated by the cGAS-STING pathway. HDAC3i can also promote the smooth synthesis of retinoic acid (RA) and the exercise of normal functions by inhibiting HDAC3, thereby achieving retinoic acid-mediated immune enhancement. Abbreviations: CDN, cyclicdinucleotide; HMGB1, high mobility group box-1 protein; HSP, heat shock protein; cGAS, cyclic GMP-AMP synthase; IFN-γ, interferon-γ; CXCL1, C-X-C motif chemokine ligand 1; CXCL2, C-X-C motif chemokine ligand 2; IL-1β, interleukin-1β; IL-6, interleukin6; BAX, Bcl-2 associated X protein; IFN-α, interferon-α; IFN-β, interferon-β; IRF3, interferon regulatory factor 3; GTP, guanosine triphosphate; ATP, adenosine triphosphate; TBK1, TANK-binding kinase 1; Caspase-3, cysteinyl aspartate specific proteinase3; ROS, reactive oxygen species; JDP2, JUN dimerization protein 2; SKi, Sloan–Kettering Institute; GPX4, glutathione peroxidase 4; GSH, glutathione; GSSG, oxidized glutathione; NrF-2, nuclear factor erythroid 2-related factor 2; HO-1, heme oxygenase-1.

**Figure 3 vaccines-13-00182-f003:**
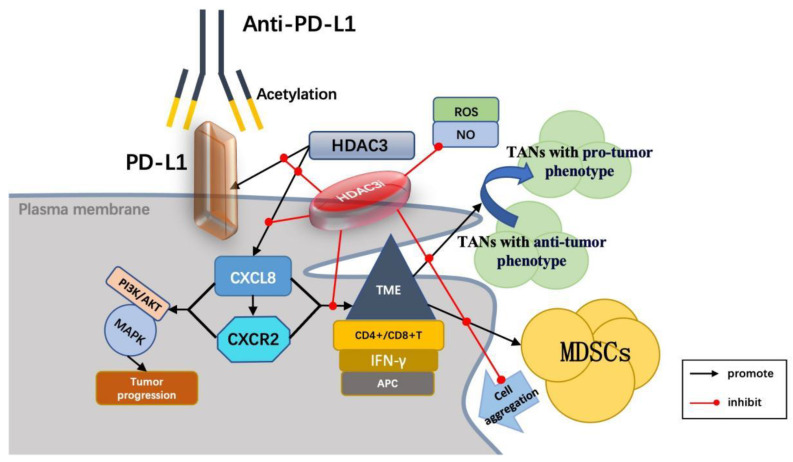
**HDAC3i promotes the transformation of tumors from “cold tumors” to “hot tumors” by blocking the CXCL8/CXCR2 axis.** HDAC3i can block the CXCL8/CXCR2 axis by inhibiting HDAC3, increasing the infiltration of CD4+/CD8+ T cells, and activating antigen-presenting cells (APCs), while preventing the recruitment of MDSCs to reshape the tumor immune microenvironment, thereby enhancing the efficacy of PD-L1 blockade therapy. Abbreviations: CXCL8, C-X-C motif chemokine ligand 8; CXCR2, C-X-C motif chemokine receptor 2; IFN-γ, interferon-γ.

**Figure 4 vaccines-13-00182-f004:**
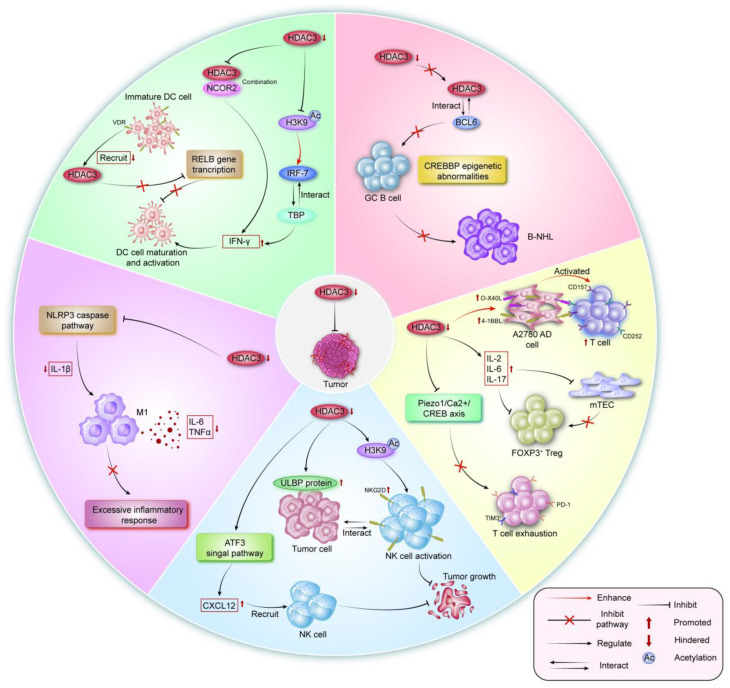
**Regulation of immune cell function by HDAC3i.** HDAC3i modulates immune responses through various mechanisms, ultimately enhancing anti-tumor immunity. By inhibiting HDAC3, these inhibitors enhance immune costimulation signals, regulate T cell subset differentiation and function, and reduce tumor immune escape. HDAC3 inhibition also restores the abnormal epigenetic programming of CREBBP, enhancing BCL6-mediated antigen presentation and IFN-γ release, which reinstates the MHC-dependent cytotoxicity of tumor-infiltrating lymphocytes and inhibits the aberrant activation of B cells and tumor progression. Additionally, HDAC3i improves NK cell anti-tumor activity by regulating ATF3 signaling, enhancing CXCL12 secretion to recruit NK cells, upregulating ULBP ligand expression, and affecting the expression of NKG2DLs. The inhibitors also promote dendritic cell maturation by upregulating IFN-γ expression, facilitating the binding of IRF7 to TBP, and inhibiting VDR recruitment of the HDAC3 complex. Furthermore, HDAC3 inhibitors reduce inflammatory factor release by regulating NLRP3/caspase-1 signaling and promote macrophage polarization toward the M2 phenotype, thereby mitigating inflammation. Abbreviations: VDR, vitamin receptor; IRF-7, interferon regulatory factor 7; H3K9, histone 3 lysine 9; NCOR2, nuclear receptor repressor 2; TBP, TATA-box binding protein; IFN-γ, interferon-γ; CREBBP, CREB-binding protein; NLRP3, NOD-like receptor protein 3; IL-1β, interleukin-1β; IL-6, interleukin-6; IL-2, interleukin-2; IL-17, interleukin-17; TNF-α, tumor necrosis factor-α; ULBP, UL16 binding protein 1; KG2-D, KG2-D type II integral membrane protein; ATF3, activating transcription factor 3; CXCL12, C-X-C motif chemokine ligand 12.

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
