# Peer review of "HDAC3: A Multifaceted Modulator in Immunotherapy Sensitization"

_vaccines, 2025, doi:10.3390/vaccines13020182_

Round 1
Reviewer 1 Report
Comments and Suggestions for Authors
Review
Title: HDAC3: A Multifaceted Modulator in Immunotherapy Sensitization
By Rui Hanet al.
This paper reviews the role of the histone deacetylase HDAC3 in cancer with the focus on its possible effect in immunotherapy.
The topic is highly relevant because epigenetic aspects of tumor adaptation to therapy, e.g. by developing resistance.
I enjoyed reading the paper: It addressed a wide spectrum aspects, among them clinics, immune regulation, immunotherapy of cancer, prevention and future direction. With 22 pages and more than 200 references the review is excessive, but, in my opinion appropriate.
The content is well structures and informative.
I have only a few optional suggestions:
1. A paragraph providing the reader with a short overview about the HDAC-superfamily and the specific role of HDAC3 (why HDAC3 is selected and, say, not HDAC2). Partly this is always addressed in the Intro, but a slight extension would involve readers not from the field.
2. The subsection 4.2 is extraordinarily long. Might the authors want to condense it a bit and/or subdivide/split it into 2 or three subtopics?
Author Response
Reviewer 1
This paper reviews the role of the histone deacetylase HDAC3 in cancer with the focus on its possible effect in immunotherapy.
The topic is highly relevant because epigenetic aspects of tumor adaptation to therapy, e.g. by developing resistance.
I enjoyed reading the paper: It addressed a wide spectrum aspects, among them clinics, immune regulation, immunotherapy of cancer, prevention and future direction. With 22 pages and more than 200 references the review is excessive, but, in my opinion appropriate.
The content is well structures and informative.
I have only a few optional suggestions:
- A paragraph providing the reader with a short overview about the HDAC-superfamily and the specific role of HDAC3 (why HDAC3 is selected and, say, not HDAC2). Partly this is always addressed in the Intro, but a slight extension would involve readers not from the field.
Answer:
We thank our reviewer for his/her constructive suggestion. We have now added a paragraph to briefly introduce the histone deacetylase (HDAC) superfamily and the specific functions of HDAC3 to improve the readability of this article,as required (line: 54-67):
Although HDAC1, HDAC2, and HDAC3 are all members of the Class I HDAC family, they perform distinct functions in gene regulation and expression. HDAC1 and HDAC2 team up with NURD, SIN3, and CoREST to form transcriptional repression complexes. They have different expression patterns and play a regulatory role in neurodevelopment and synaptic plasticity, with higher expression in actively proliferating cells. On the contrary, HDAC3 forms inhibitory complexes with NCOR or SMRT. These complexes are widely present in metabolic tissues and are crucial for regulating and reshaping metabolic activities and inflammatory responses. Since tumor occurrence is closely linked to abnormal metabolic activities and unpredictable strong inflammatory responses, HDAC3 has attracted increasing exploration and attention as a target for tumor treatment strategies[9]. In particular, our published study has primarily investigated the multiple anti-cancer effects of HDAC3 inhibition treatment, which also shows great potential in enhancing the sensitivity of cancer immunotherapy[10, 11].
- The subsection 4.2 is extraordinarily long. Might the authors want to condense it a bit and/or subdivide/split it into 2 or three subtopics?
Answer:
Many thanks to our reviewer for such considering advise. We have now properly condensed the content and disassembled the 4.2 chapters into three parts according to the logical relationship (line: 713-839 ):
4.2.Inhibition of HDAC3 for improving anti-cancer immunity
Tumor immunotherapy differs from conventional therapies, including surgery, radiotherapy, and chemotherapy. Its primary objective is to enhance and reactivate the host's anti-tumor immune response by increasing the immunogenicity of tumor cells and stimulating the autoimmune system,and this approach has clinical advantages such as durability, thoroughness, and comprehensiveness. Although the drug response rate remains limited, immunotherapy continues to provide clinical benefits for the majority of cancer patients. The ecology of tumor immunity is inherently complex. Under normal circumstances, the immune system recognizes and eliminates tumor cells within the host. However, tumor cells have evolved various strategies and mechanisms to evade this immune response, including the recruitment of immunosuppressive cells, the impairment of antigen presentation, and the induction of effector cell dysfunction. This result contributed to tumor survival during various stages of anti-tumor immunotherapy[173].
4.2.1. Existing antitumor immunotherapy
Understanding the underlying mechanisms of interactions between targets and cells within the tumor microenvironment is crucial for the development of new immunotherapies and the enhancement of treatment response rates. Currently, tumor immunotherapy, grounded in progressively mature mechanistic theories and technologies, has achieved significant advancements in both basic and clinical research. Tumor immunotherapy, represented by oncolytic virus therapy, tumor vaccines, adoptive cell transfer therapy, immune checkpoint inhibitors, and cytokine therapy, has become a hot topic in the field of cancer treatment[174, 175]. Oncolytic virus therapy represents one of the earliest forms of immunotherapy, using modified viruses to target cancer cells. When these viruses infect tumors, they trigger inflammation and activate the body's immune defenses against cancer. T-Vec, a modified herpes simplex virus, has proven particularly effective against advanced melanoma that can't be surgically removed[176]. Cancer vaccines work by introducing inactive tumor antigens to stimulate T cell responses against cancer. The melanoma antigen GP100, for example, has successfully activated tumor-fighting immune cells. In prostate cancer treatment, Provenge stands out as the first FDA-approved immunotherapy vaccine. This treatment uses the patient's own immune cells combined with the PA2024 antigen to fight metastatic castration-resistant prostate cancer[177][178]. By collecting the patient's peripheral blood mononuclear cells and co-culturing them with the antigen PA2024 in vitro, and subsequently infusing these cells back into the patient, dendritic cells (DCs) can be effectively induced to mature and activate T cell-mediated immune responses that identify and eliminate prostate cancer cells expressing the PAP antigen. These findings underscore the unique role of cancer vaccines in cancer treatment[179]. Immune checkpoint inhibitors (ICI) have revolutionized cancer treatment by targeting natural immune system regulators. These checkpoints normally prevent autoimmune responses, but cancer cells exploit them to hide from immune attack. Two key checkpoints, PD-1 and CTLA-4, act like brakes on the immune system. CTLA-4, found on regulatory T cells, suppresses immune responses by blocking the stimulatory signals that normally occur when B7-1/2 molecules bind to CD28 receptors on T cells[180]. Several immune checkpoint inhibitors have proven highly effective in cancer treatment. These include Ipilimumab, which targets CTLA-4; Nivolumab and Pembrolizumab, which block PD-1; and Avelumab, Durvalumab, and Atezolizumab, which target PD-L1. Ipilimumab, the first approved checkpoint inhibitor, works by preventing CTLA-4 from binding to its receptor, allowing T cells to mount a sustained immune response. Similarly, Nivolumab disrupts the interaction between PD-1 and PD-L1, preventing tumors from evading immune detection and restoring T cells' ability to attack cancer cells. Research continues to uncover new immune checkpoints, including TIGIT, B7-H6, TIM-3, LAG-3, BTLA, and IDO2. Scientists are actively studying these targets to develop the next generation of checkpoint inhibitors[181]. Adoptive cell transfer therapy harnesses the body's own immune cells to fight cancer. The process begins by collecting immune cells - either from the patient's blood, tumor tissue, or from donors. These cells are then activated and multiplied in the laboratory using compounds like IL-2 and anti-CD3 antibodies before being returned to the patient. Two advanced forms of this approach have shown remarkable promise. CAR-T therapy engineers T cells to carry specialized receptors that recognize specific tumor markers, enabling them to release powerful immune signals and precisely target cancer cells. TCR-T therapy represents another breakthrough in this field, adding to the growing arsenal of cellular immunotherapies.These techniques have proven particularly effective in cancer treatment, marking a significant advance in personalized immunotherapy[182, 183]. TCR engineering of tumor-infiltrating lymphocytes (TILs) represents one of the most promising treatments for tumors, achieved by transducing chimeric antigen receptors or TCRα/β heterodimers into TILs to enhance their specific binding affinity to tumor-associated antigens (TAAs). Kimmtrak (tebentafusp-tebn), the first bispecific TCR-T therapy approved by the FDA, has demonstrated breakthrough results in the treatment of unresectable or metastatic uveal melanoma (mUM)[184]. Similarly, satisfactory anti-tumor efficacy was clinically observed after TCR genes were transferred to peripheral blood lymphocytes (PBLs) of melanoma patients[185].Cytokine therapy fights cancer by using immune-stimulating proteins like IL-2 and IFN-α to activate and multiply immune cells. These cytokines act as cellular messengers, promoting dendritic cell maturation and enhancing T cells' ability to kill cancer cells. To minimize side effects and improve patient tolerance, cytokines are often used alongside cell transplant therapies[186]. Immunotherapy has become a cornerstone of modern cancer treatment, with T cells playing the central role. The approach focuses on four key strategies: triggering cytokine release, activating T cells' cancer-fighting abilities, preventing T cell exhaustion, and introducing specially engineered T cells that can recognize and target cancer cells.
4.2.2. Challenges and obstacles of antitumor immunotherapy
Although anti-tumor immunotherapy has made significant progress in recent years, several deficiencies and challenges continue to hinder its application in clinical practice. A primary challenge is the low response rate; for instance, anti-PD-1/PD-L1 therapy is effective only for a subset of patients, with response rates typically ranging from 20% to 40%. Consequently, many patients do not derive any benefit from this treatment.This phenomenon is attributed not only to significant individual differences among patients but also to the high heterogeneity observed in certain tumor cells, which can render immunotherapy ineffective for specific tumor subtypes. Additionally, some tumors exhibit low immunogenicity, commonly referred to as "cold tumors," and the physical isolation provided by the extracellular matrix can impede the infiltration of immune cells. In tumor microenvironments, there exists a significant presence of immunosuppressive cells, including Tregs and MDSCs, along with elevated levels of immunosuppressive cytokines and metabolic products. This collaborative effect weakens the immune response, making it challenging for the immune system to identify and attack tumor cells. Under certain conditions, this tumor immunosuppressive microenvironment can even induce primary or acquired drug resistance in patients undergoing immunotherapy. These factors complicate the exploration and development of effective immunotherapy strategies.
4.2.3. HDAC3 inhibitors have the potential to become sensitizers of immunotherapy
Tumor immune escape is a major challenge currently faced in tumor immunotherapy. Tumor cells can secrete immunosuppressive factors and induce the generation of immunosuppressive cells, or they may rely on ligand pathways such as FAS/FAS-L and PD1/PDL1 to activate and amplify immunosuppressive signals,this enables them to evade recognition and attack by the host's immune cells[187]. Research on HDAC3 inhibitors as sensitizers for tumor treatment has been reported. In an in vitro experiment, the knockout of HDAC3 using CRISPR/Cas9 resulted in increased DNA damage in irradiated tumor cells, thereby reversing the resistance of rhabdomyosarcoma (RMS) to radiotherapy and exerting a radiosensitization effect[2]. Additionally, one study demonstrated that the use of HDAC3 inhibitors promoted AKT protein acetylation, which enhanced the sensitivity of leukemia cells to chemotherapy drugs[188].It is exciting that the current research shows that the HDAC inhibitor has gradually shown good potential in the effect of sensitizing tumor immunotherapy.
In gastric cancer cell models, the knockdown of HDAC3 inhibited the nuclear translocation of STAT1, impaired IFN-γ signaling, reduced B7-H1 expression in HGC-27 cells. Ultimately, the percentage of infiltrating CD8+ T cells in the tumor microenvironment is increased and the immune evasion of tumor cells is inhibited[189]. In addition, in the B16 melanoma cell model, treatment with AR42 (a histone deacetylase inhibitor) resulted in a reduction of PD-L1 and PD-L2 expression in B16 cells, while simultaneously increasing the infiltration of neutrophils and natural killer (NK) cells Within the tumor microenvironment (TME), enhances the body’s immune response to tumor cells[44]. Research using a mouse fibrosarcoma model shows that HDAC3 suppresses the CXCL9/10/11-CXCR3 pathway by binding to chemokine promoters. When HDAC3 is blocked, it slows the growth of MCA205 tumors and draws more CXCR3+ T-cells into the tumor environment. This leads to increased numbers of CD4+IFNγ+, CD8+IFNγ+, and CD11b+F4/80+ immune cells in the tumors, ultimately helping to destroy cancer cells[190]. Therefore, the above evidence indicates that targeting HDAC3 is an effective strategy that may enhance anti-tumor immunity.
The above studies have indicated that inhibiting the expression of HDAC3 can significantly enhance the functionality of immune cells, particularly T cells. This inhibition promotes T cell-mediated anti-tumor immune responses by upregulating costimulatory signals such as OX-40L and 4-1BBL, facilitating the release of IL-2, and reducing the expression of immune checkpoints. Additionally, this inhibition impacts other immune cells, including NK and dendritic cells (DC), exerting a positive stimulatory effect. Studies have shown that class I HDAC inhibitors (HDACi), specifically Chidamide, enhance the persistence and anti-tumor effects of CAR-T cells by upregulating the expression of transcription factors such as LEF1 and TCF4, as well as activating the Wnt/β-catenin pathway[192]. One study demonstrated nanomicelles that encapsulate siRNA-PD-L1 and combined them with HDAC inhibitors (HDACi) to effectively reverse T cell exhaustion and prevent the immune evasion of tumor cells. This discovery highlights the potential of epigenetic regulatory drugs and underscores the promising prospects of combining PD-1/PD-L1 blocking therapy[193]. Based on the evidence presented, while immunotherapy has achieved significant breakthroughs in clinical treatment, it still faces obstacles and challenges, including low response rates and unpredictable potential side effects. Nevertheless, both in terms of mechanisms of action and in light of existing research findings, there are compelling reasons to believe that HDAC3 inhibitors may serve as sensitizers when combined with immunotherapy, thereby enhancing the efficacy of immunotherapeutic approaches, and underscores the clinical value of tumor immunotherapy and facilitates the exploration and development of novel clinical treatment strategies.

Reviewer 2 Report
Comments and Suggestions for Authors
The authors describe the HDAC3's multifaceted roles, focusing on its regulation of key immune-modulatory pathways such as cGAS-STING, ferroptosis, and the Nrf2- HO-1 axis. Furthermore, they explore its influence on tumor biology, as a potential biomarker for cancer prognosis, as well as a therapeutic target. The manuscript is well-written and clear regarding several associations of HDAC3 with the immune regulation, immune signaling pathways, Retinoic acid (RA), T Lymphocytes and macrophages. I recommend the manuscript without changes to do.
Author Response
Reviewer 2
The authors describe the HDAC3's multifaceted roles, focusing on its regulation of key immune-modulatory pathways such as cGAS-STING, ferroptosis, and the Nrf2- HO-1 axis. Furthermore, they explore its influence on tumor biology, as a potential biomarker for cancer prognosis, as well as a therapeutic target. The manuscript is well-written and clear regarding several associations of HDAC3 with the immune regulation, immune signaling pathways, Retinoic acid (RA), T Lymphocytes and macrophages. I recommend the manuscript without changes to do.
Answer:
We sincerely appreciate our reviewer for his/her professional and constructive feedback, as well as your diligent efforts and endorsement throughout the review process. I believe that your commendation will enable more colleagues in the field of tumor research to recognize this result. We have further optimized the article to enhance the readability for readers.

Reviewer 3 Report
Comments and Suggestions for Authors
The manuscript “HDAC3: A Multifaceted Modulator in Immunotherapy Sensitization” by Rui Han et al. is dedicated to reviewing HDAC3 as a target for immunotherapy of malignancies. The manuscript is well structured, written in clear scientific language and can certainly be published after revision.
The following should be noted as comments:
1. Lines 26-28. “...on the clinical application of HDAC3...”. In my opinion, this statement is formulated incorrectly, it should be used: “..the use of HDAC3 inhibitors...”.
2. There is a critical remark to the "Introduction" section as a whole. This manuscript is primarily devoted specifically to HDAC3, while a fairly small amount of information is devoted to the role of HDAC and HDACi. I would like to recommend that the authors slightly revise the structure of the “Introduction”. At the beginning of this section, 2-3 paragraphs should be added about HDAC and HDACi in general, without reference to a specific isoform, emphasizing their critical importance in the development of a wide range of diseases and the active search for therapeutic strategies for their treatment. In this way, the authors will be able to focus on the relevance of their review work and the need for its publication. I would like to convincingly recommend that the authors cite information from the following sources, adding them to the list of references (currently, a fairly extensive work presented on 22 pages includes a somewhat limited number of sources, and therefore it would be recommended to expand the list of references with links to relevant sources):
- HDACs and HDAC Inhibitors in Cancer Development and Therapy. Cold Spring Harbor perspectives in medicine vol. 6,10 a026831. 3 Oct. 2016, doi:10.1101/cshperspect.a026831
- Hydroxamic Acids Containing a Bicyclic Pinane Backbone as Epigenetic and Metabolic Regulators: Synergizing Agents to Overcome Cisplatin Resistance. Cancers vol. 15,20 4985. 14 Oct. 2023, doi:10.3390/cancers15204985
- Restoring the epigenome in Alzheimer's disease: advancing HDAC inhibitors as therapeutic agents. Drug discovery today vol. 29,7 (2024): 104052. doi:10.1016/j.drudis.2024.104052
- Elaboration of the Effective Multi-Target Therapeutic Platform for the Treatment of Alzheimer's Disease Based on Novel Monoterpene-Derived Hydroxamic Acids. International journal of molecular sciences vol. 24,11 9743. 4 Jun. 2023, doi:10.3390/ijms24119743
- HDAC inhibitors as pharmacological treatment for Duchenne muscular dystrophy: a discovery journey from bench to patients. Trends in molecular medicine vol. 30,3 (2024): 278-294. doi:10.1016/j.molmed.2024.01.007
- New Spirocyclic Hydroxamic Acids as Effective Antiproliferative Agents. Anti-cancer agents in medicinal chemistry vol. 21,5 (2021): 597-610. doi:10.2174/1871520620666200527132420
- Advances and Challenges of HDAC Inhibitors in Cancer Therapeutics. Advances in cancer research vol. 138 (2018): 183-211. doi:10.1016/bs.acr.2018.02.006
- Recent developments of HDAC inhibitors: Emerging indications and novel molecules. British journal of clinical pharmacology vol. 87,12 (2021): 4577-4597. doi:10.1111/bcp.14889
- Novel Multitarget Hydroxamic Acids with a Natural Origin CAP Group against Alzheimer's Disease: Synthesis, Docking and Biological Evaluation. Pharmaceutics vol. 13,11 1893. 8 Nov. 2021, doi:10.3390/pharmaceutics13111893
- Epigenetic modulation and understanding of HDAC inhibitors in cancer therapy. Life sciences vol. 277 (2021): 119504. doi:10.1016/j.lfs.2021.119504
3. Line 354. The authors should check the correctness of the use of the abbreviation: histone acetylene 3 (HDAC3), identical to histone deacetylase 3 (HDAC3).
4. In my opinion, paragraph 3.1.6 is written incorrectly and is a confusing, inconsistent text. The authors should pay some attention to this section so that readers do not have the feeling of reading an incomprehensible text.
In conclusion, I would like to thank the team of authors for their research and their desire to present it to the scientific community, as well as wish them success in their future work.
Author Response
Reviewer 3
The manuscript “HDAC3: A Multifaceted Modulator in Immunotherapy Sensitization” by Rui Han et al. is dedicated to reviewing HDAC3 as a target for immunotherapy of malignancies. The manuscript is well structured, written in clear scientific language and can certainly be published after revision.
The following should be noted as comments:
- Lines 26-28. “...on the clinical application of HDAC3...”. In my opinion, this statement is formulated incorrectly, it should be used: “..the use of HDAC3 inhibitors...”.
Answer:
We thank our reviewer for his/her professional comments for improving the quality of this manuscript. We have revised the sentence as “this study provides a forward-looking perspective on the clinical application of HDAC3 inhibitors” (line: 27-28).
- There is a critical remark to the "Introduction" section as a whole. This manuscript is primarily devoted specifically to HDAC3, while a fairly small amount of information is devoted to the role of HDAC and HDACi. I would like to recommend that the authors slightly revise the structure of the “Introduction”. At the beginning of this section, 2-3 paragraphs should be added about HDAC and HDACi in general, without reference to a specific isoform, emphasizing their critical importance in the development of a wide range of diseases and the active search for therapeutic strategies for their treatment. In this way, the authors will be able to focus on the relevance of their review work and the need for its publication. I would like to convincingly recommend that the authors cite information from the following sources, adding them to the list of references (currently, a fairly extensive work presented on 22 pages includes a somewhat limited number of sources, and therefore it would be recommended to expand the list of references with links to relevant sources):
- HDACs and HDAC Inhibitors in Cancer Development and Therapy. Cold Spring Harbor perspectives in medicine vol. 6,10 a026831. 3 Oct. 2016, doi:10.1101/cshperspect.a026831
- Hydroxamic Acids Containing a Bicyclic Pinane Backbone as Epigenetic and Metabolic Regulators: Synergizing Agents to Overcome Cisplatin Resistance. Cancers vol. 15,20 4985. 14 Oct. 2023, doi:10.3390/cancers15204985
- Restoring the epigenome in Alzheimer's disease: advancing HDAC inhibitors as therapeutic agents. Drug discovery today vol. 29,7 (2024): 104052. doi:10.1016/j.drudis.2024.104052
- Elaboration of the Effective Multi-Target Therapeutic Platform for the Treatment of Alzheimer's Disease Based on Novel Monoterpene-Derived Hydroxamic Acids. International journal of molecular sciences vol. 24,11 9743. 4 Jun. 2023, doi:10.3390/ijms24119743
- HDAC inhibitors as pharmacological treatment for Duchenne muscular dystrophy: a discovery journey from bench to patients. Trends in molecular medicine vol. 30,3 (2024): 278-294. doi:10.1016/j.molmed.2024.01.007
- New Spirocyclic Hydroxamic Acids as Effective Antiproliferative Agents. Anti-cancer agents in medicinal chemistry vol. 21,5 (2021): 597-610. doi:10.2174/1871520620666200527132420
- Advances and Challenges of HDAC Inhibitors in Cancer Therapeutics. Advances in cancer research vol. 138 (2018): 183-211. doi:10.1016/bs.acr.2018.02.006
- Recent developments of HDAC inhibitors: Emerging indications and novel molecules. British journal of clinical pharmacology vol. 87,12 (2021): 4577-4597. doi:10.1111/bcp.14889
- Novel Multitarget Hydroxamic Acids with a Natural Origin CAP Group against Alzheimer's Disease: Synthesis, Docking and Biological Evaluation. Pharmaceutics vol. 13,11 1893. 8 Nov. 2021, doi:10.3390/pharmaceutics13111893
- Epigenetic modulation and understanding of HDAC inhibitors in cancer therapy. Life sciences vol. 277 (2021): 119504. doi:10.1016/j.lfs.2021.119504
Answer:
We thank our reviewer for such great suggestion. After carefully reading the recommended literatures, we selected several articles that are highly relevant to the content of this article and cited them in line 40-51.
In addition, we have revised and improved the section of “Introduction” by adding following content:
A: Existing evidence indicates that HDACs play an essential role in the progression of various diseases. One study demonstrated that a series of novel single-pyrene-derived hydroxyl acids significantly inhibit HDAC activity, thereby reducing the production of Aβ and decreasing tau protein phosphorylation, which helps restore cognitive function impaired by Alzheimer's disease. Based on their physiological and pathological effects, the HDAC family can be classified into four categories: I, II, III, and IV. Notably, the HDAC family often exhibits abnormally high expression in tumors, promoting tumor growth, invasion, and metastasis. Various types of HDAC inhibitors are currently under gradual development, with a particular focus on exploring and developing selective HDAC inhibitors to optimize the absorption, distribution, metabolism, and excretion characteristics of these compounds while minimizing toxicity and side effects[4-6].( line:40-51)
B: Although HDAC1, HDAC2, and HDAC3 are all members of the Class I HDAC family, they perform distinct functions in gene regulation and expression. HDAC1 and HDAC2 team up with NURD, SIN3, and CoREST to form transcriptional repression complexes. They have different expression patterns and play a regulatory role in neurodevelopment and synaptic plasticity, with higher expression in actively proliferating cells. On the contrary, HDAC3 forms inhibitory complexes with NCOR or SMRT. These complexes are widely present in metabolic tissues and are crucial for regulating and reshaping metabolic activities and inflammatory responses. Since tumor occurrence is closely linked to abnormal metabolic activities and unpredictable strong inflammatory responses, HDAC3 has attracted increasing exploration and attention as a target for tumor treatment strategies[9]. In particular, our published study has primarily investigated the multiple anti-cancer effects of HDAC3 inhibition treatment, which also shows great potential in enhancing the sensitivity of cancer immunotherapy[10, 11]. ( lines54-67 )
- Line 354. The authors should check the correctness of the use of the abbreviation: histone acetylene 3 (HDAC3), identical to histone deacetylase 3 (HDAC3).
Answer:
We appreciate the reviewer's meticulous review. We have checked and revised all the correctness of the use of the abbreviation in our manuscript, including the one pointed out by our reviewer in line 354 (now 378).
- In my opinion, paragraph 3.1.6 is written incorrectly and is a confusing, inconsistent text. The authors should pay some attention to this section so that readers do not have the feeling of reading an incomprehensible text.
Answer:
We thank our reviewer for his/her professional suggestion. Section 3.1 aims to discuss the relationship between HDAC3 and immune signaling pathways, and Section 3.1.5 is used to introduce the relationship between HDAC3 and the immune regulation mechanism related to RA. The section titles in our previous version were inappropriate, which affected the readability and caused confusion. We have optimized the section titles and adjusted the section content to enhance the readability and logic of the article. We have merged the original Sections 3.1.5 and 3.1.6 into one section and named it "HDAC3 in Regulating Retinoic Acid (RA) Signals for Immune Regulation"( line: 396-462):
3.1.5. HDAC3 in Regulating Retinoic Acid (RA) Signals for Immune Regulation
Retinoic acid (RA) is the primary bioactive metabolite of retinol (vitamin A). It plays a significant regulatory role in both innate and adaptive immunity by influencing the growth, differentiation, and migration of immune cells. The immune effects mediated by RA depend on the specific cell type and the site of RA release within the immune microenvironment. In the body, retinoic acid exists in various forms, such as 9-cis and all-trans, and interacts with retinoic acid receptors (RAR) to regulate the transcription of downstream target genes[116]. Adapalene, an agonist of the retinoic acid receptor (RAR), effectively promotes the cellular senescence-associated secretory phenotype (SASP) and enhances the tumor clearance of prostate cancer by natural killer (NK) cells[117]. Evidence suggests that retinoic acid (RA) plays a critical role in the immune regulation of the organism. For instance, RA promotes the development and maturation of fetal lymphoid tissue inducer (LTi) cells during pregnancy, thereby enhancing the offspring's immune response. Additionally, RA regulates the balance between Th17 and Treg cells in accordance with the conditions and status of the immune microenvironment, thereby maintaining immune homeostasis[118, 119].
Retinoic acid (RA) also is important cofactor for the activation and enhancement of humoral immunity, effectively promotes the renewal and maturation of B lymphocytes and induces their differentiation into plasma cells. RA significantly up-regulates the expression of B lymphocyte-induced maturation protein 1 (Blimp-1) in germinal center (GC) B lymphocytes and activates the expression of activation-induced cytidine deaminase (AID) in these cells, thereby supporting the normal release of immunoglobulin G (IgG)[120-122]. Furthermore, RA signaling activates follicular dendritic cells (FDCs) via retinoic acid receptors (RAR) and up-regulates the expression of chemokine ligand 13 (CXCL13) and B cell activation factor (BAFF), processes that facilitate B cell survival and migration in the gut and enhance the immune response[123-125].
Similarly, the percentage of NK cells in the peripheral circulatory system is positively correlated with the level of RA, which upregulates the expression of MHC class I chain-related molecules A (MICA) in tumor cells,MICA initiates the NK cell immune response by binding to the natural killer cell group 2D (NKG2D) receptor on the surface of NK cells[126].
The relationship between intestinal mucosal immune dysregulation and the development of intestinal malignant tumors has garnered increasing attention and research. The excessive release of a variety of pro-inflammatory factors will induce the occurrence of inflammatory bowel disease, seriously damaging the integrity of the intestinal barrier and the homeostasis of intestinal mucosal immunity. All-trans retinoic acid (atRA) inhibited the release of inflammatory mediators, including TNF-α, NO, IL-12, PGE2, and COX-2, in lipopolysaccharide (LPS)-activated macrophages, thereby inhibiting the occurrence of excessive inflammatory responses[127]. RA induces the differentiation of DC precursors (pre-DC) to CD103+CD11b+DC via the intestinal transit receptor α4β7, and this subtype of DC mediates the generation of Foxp3+ regulatory T cells and IL-10-producing effector T cells to alleviate pathogen-induced intestinal inflammation. In addition, CD103+CD11b+ DC synthesis releases retinoic acid into the peripheral circulatory system, which mediates intestinal homing of immune effector cells by inducing target cells to synthesise and express α4β7 and chemokine receptor 9 (CCR9) on the cell membrane surface, and ultimately through the α4β7/MAdCAM-1 and CCR9/CCL25 axis. During the inflammatory response phase, RA induces DCs to produce inflammatory mediators and promotes the differentiation of effector T cells, while inducing the formation of,Tertiary lymphoid structures (TLS) and driving an adaptive immune response. These physiological processes mediated by retinoic acid are essential for the maintenance of intestinal immune homeostasis[128-131].
There is still more evidence that Retinoic acid plays a non-negligible role in activating and maintaining immune responses[132].Existing evidence shows that HDAC3 achieves immune regulation by affecting the transformation of the RA signal.HDAC3 negatively regulates the synthesis and function of retinoic acid, while the transcription factor JDP2 inhibits retinoic acid (RA)-dependent transcription by recruiting the histone deacetylase 3 (HDAC3) complex to the promoter regions of target genes. Additionally, JDP2 inhibits the activation of the p300/ATF-2 axis through the recruitment of HDAC3. This recruitment results in a decrease in RA-induced c-jun gene transcription and cell differentiation[133-135]. Additionally, butyric acid, a short-chain fatty acid (SCFA), enhances RA expression in both human and mouse epithelial cells through the inhibition of HDAC3. Similarly, the application of HDAC3 inhibitors has been shown to produce comparable effects[136]. Ski protein is a negative regulator of retinoic acid (RA) signaling. Ski inhibits retinoic acid (RA) signaling by interacting with key components of HDAC3 and acting as a transcriptional corepressor. Therefore, silencing HDAC3 or using HDAC3 inhibitors may facilitate the synthesis and normal function of retinoic acid, thereby exerting beneficial effects on maintaining and enhancing anti-tumor immune responses[137, 138].(Figure2).
In conclusion, I would like to thank the team of authors for their research and their desire to present it to the scientific community, as well as wish them success in their future work.
Answer:
We are truly grateful to our reviewer for providing professional and constructive feedback, along with diligent efforts and unwavering support throughout the entire review process. We are confident that your endorsement will help gain recognition for this result among a wider audience of colleagues in the field of tumor research.
